# Methane emissions from a Californian landfill, determined from airborne remote sensing and in-situ measurements

Sven Krautwurst[1], Konstantin Gerilowski[1], Haflidi H. Jonsson[2], David R. Thompson[3], Richard W. Kolyer[4], Laura T. Iraci[4], Andrew K. Thorpe[3], Markus Horstjann[1], Michael Eastwood[3], Ira Leifer[5], Sam Vigil[6], Thomas Krings[1], Jakob Borchardt[1], Michael Buchwitz[1], Matthew M. Fladeland[4], John P. Burrows[1], and Heinrich Bovensmann[1]

[1]Institute of Environmental Physics (IUP), University of Bremen, Bremen, Germany
[2]Center for Interdisciplinary Remotely-Piloted Aircraft Studies (CIRPAS), Marina, CA, US
[3]Jet Propulsion Laboratory (JPL), California Institute of Technology (Caltech), Pasadena, CA, US
[4]Earth Science Devision, NASA Ames Research Center (ARC), Mountain View, CA, US
[5]Bubbleology Research International (BRI), Goleta, CA, US
[6]California Polytechnic State University (CalPoly), San Luis Obispo, CA, US

*Correspondence to:* Sven Krautwurst (krautwurst@iup.physik.uni-bremen.de)

**Abstract.** Fugitive emissions from waste disposal sites are important anthropogenic sources of the greenhouse gas methane ($CH_4$). As a result of the growing world population and the recognition of the need to control greenhouse gas emissions, this anthropogenic source of $CH_4$ has received much recent attention. However, the accurate assessment of the $CH_4$ emissions from landfills by modeling and existing measurement techniques is challenging. This is because of inaccurate knowledge of the model parameters and the extent of and limited accessibility to landfill sites. This results in a large uncertainty in our knowledge of the emissions of $CH_4$ from landfills and waste management.

In this study, we present results derived from data collected during the research campaign COMEX ($CO_2$ and Methane EXperiement) in late summer 2014 in the Los Angeles (LA) Basin. One objective of COMEX, which comprised aircraft observations of methane by the remote sensing Methane Airborne MAPper (MAMAP) instrument and a Picarro greenhouse gas in-situ analyser, was the quantitative investigation of $CH_4$ emissions.

Enhanced $CH_4$ concentrations or „$CH_4$ plumes" were detected downwind of landfills by remote sensing aircraft surveys. Subsequent to each remote sensing survey, the detected plume was sampled within the atmospheric boundary layer by in-situ measurements of atmospheric parameters such as wind information and dry gas mixing ratios of $CH_4$ and carbon dioxide ($CO_2$) from the same aircraft. This was undertaken to facilitate the independent estimation of the surface fluxes for the validation of the remote sensing estimates.

During the COMEX campaign, four landfills in the LA Basin were surveyed. One landfill repeatedly showed a clear emission plume. This landfill, the Olinda Alpha Landfill, was investigated on four days during the last week of August and first days of September 2014. Emissions were estimated for all days using a mass balance approach. The derived emissions vary between 11.6 and $17.8 \, \mathrm{kt} \, CH_4 \, \mathrm{yr}^{-1}$ with related uncertainties in the range of $14\,\%$ to $45\,\%$. The comparison of the remote sensing and in-situ based $CH_4$ emission rate estimates reveals good agreement within the error bars with an average of the absolute differences of around $2.4 \, \mathrm{kt} \, CH_4 \, \mathrm{yr}^{-1}$ ($\pm 2.8 \, \mathrm{kt} \, CH_4 \, \mathrm{yr}^{-1}$). The US Environmental Protection Agency (EPA) reported inven-

tory value is $11.5\,\mathrm{kt\,CH_4\,yr^{-1}}$ for 2014, on average $2.8\,\mathrm{kt\,CH_4\,yr^{-1}}$ ($\pm 1.6\,\mathrm{kt\,CH_4\,yr^{-1}}$) lower than our estimates acquired in the afternoon in late Summer 2014. This difference may in part be explained by a possible leak located on the south-western slope of the landfill, which we identified in the observations of the Airborne Visible / Infrared Imaging Spectrometer - Next Generation (AVIRIS-NG) instrument, flown contemporaneously aboard a second aircraft on one day.

## 1 Introduction

Methane ($CH_4$) is one of the most important anthropogenic greenhouse gases modulated by human activity. According to Saunois et al. (2016), the methane emissions from landfills and waste management contribute with around $15\,\%$ to $18\,\%$ to the global anthropogenic methane emissions budget. Under anaerobic conditions, bacteria produce $CH_4$ by consuming biodegradable waste, which has been deposited within the landfill. This is known as landfill gas (LFG), which contains $CH_4$ as its major component (typically between $50\,\%$ to $60\,\%$), carbon dioxide ($CO_2$) and other gases (e.g., Eklund et al., 1998; Amini et al., 2012).

Modern landfills (NSWMA, 2006) are often covered with special oxidation layers and are also equipped with tubes embedded vertically and / or horizontally within the landfill, through which the LFG is collected. The collected LFG is often used (and converted to $CO_2$) in small dedicated power plants for electricity and heat generation and, thus, reduces the environmental impact of the landfill emissions. When not used for power generation, collected LFG is sometimes flared, which also oxidises $CH_4$ to $CO_2$ having a lower global warming potential (Myhre et al., 2013).

Nevertheless, not all of the $CH_4$ is captured by the LFG collection system and subsequently converted to $CO_2$. The amount of the remaining $CH_4$ escaping into the atmosphere depends on the engineering approaches used to manage the landfill and atmospheric boundary layer conditions. For instance, the type and material of the landfill cover can decrease (Trapani et al., 2013) or increase emissions (Capaccioni et al., 2011). Trapani et al. (2013) have also found that slopes of landfills are areas with an enhanced $CH_4$ release. Additionally, atmospheric pressure variations (Czepiel et al., 2003; Poulsen et al., 2003; Gebert and Groengroeft, 2006; Trapani et al., 2013; Xu et al., 2014) or surface wind speeds (Poulsen and Moldrup, 2006) can modulate $CH_4$ emissions into the atmosphere.

Both measurements of $CH_4$ and models of the processes producing $CH_4$ in landfill sites can be used to estimate their emissions. Commonly used for reporting and recommended by IPCC (2006) are first-order decay (FOD) waste models. They are based on knowledge of the amount of available degradable waste, which is consumed by the bacteria, how it decays over time, but also consider other parameters such as the type and age of the waste, its temperature, moisture content and oxidation capacity of the landfill cover (Amini et al., 2013). Studies comparing direct measurements to model estimates found that the modeled outputs can significantly differ from actual measurements (Amini et al., 2012, 2013).

However, measurements are also challenging because landfills typically have a relatively large surface area (up to some square kilometres), an irregular topography and the emissions are not distributed homogeneously across the landfill. Babilotte et al. (2010) compared five different techniques measuring emissions from the same landfill. The study included ground based in-situ (tracer gas method, inverse modeling of direct $CH_4$ measurements), ground based remote sensing (laser plume mapping,

differential absorption light detection and ranging), and an airborne based method (helicopter-borne infrared laser spectroscopy at around 1.65 μm). The $CH_4$ emission rate estimates of the landfill under consideration and of a controlled release experiment performed in that study disagree by a factor of 5 to 10.

Several other studies used airborne based in-situ measurements to characterize the total emissions of landfills (e.g., Peischl et al., 2013; Lavoie et al., 2015, and references therein). In these studies different flight strategies and mass balance approaches were used. Emission uncertainties are typically estimated to be between approximately 20 % and 30 %. However, airborne in-situ measurements are often restricted by Air Traffic Control (ATC) regulations such as minimum safe altitude and ATC control zones.

Recently, airborne thermal-infrared (TIR, 7.5 to 13.5 μm) imaging spectrometry measurements were tested to locate $CH_4$ emissions also from landfills (Tratt et al., 2014). The study succeeded to derive emission rates for two localized on-site emitters - a compressed natural gas (CNG) fueling station and a gas-flaring station - with relative errors of 50 % and 120 %. However, integrated emissions for the entire landfill are not reported.

In this manuscript, we present a data set collected by two different techniques i.e. passive airborne remote sensing and airborne in-situ cavity ring down spectroscopy (CRDS). They were used to investigate the ability of remote-sensing measurements to determine emission rates and to independently estimate the emission rate of a particular landfill in the Los Angeles (LA) Basin on four different days in late summer 2014. The passive airborne remote sensing method is based on medium spectral resolution (~0.9 nm) solar absorption spectroscopy in the short wave infrared (SWIR) region around 1.65 μm. To assess total $CH_4$ landfill emissions, a mass balance approach was used. The emission estimates, derived by this method, were compared to emission estimates calculated using airborne in-situ measurements acquired from the same aircraft. Emissions were estimated using a Kriging method for interpolation of the data in combination with a mass balance approach (in a similar way as described in, e.g., Lavoie et al., 2015, and references therein). In addition, imaging spectroscopy observations from another passive remote sensing instrument installed aboard a second aircraft were utilized to identify emission hotspots across the landfill by analysing measured spectra in the region around 2.3 μm at low spectral resolution (~5 nm).

This article is structured as follows: Section 2 introduces the investigated targets, participating instruments and the applied flight strategy. The retrieval methods are described in Sect. 3. In Sect. 4, the main results of this study are presented. This includes the estimated emission rates, their errors, the retrieval results from the imaging instrument aboard the second aircraft and also comparisons between the instruments and reported inventory estimates. The manuscript closes with a summary and conclusions (Sect. 5).

## 2 Measurements

This section gives a short description of the research campaign in which the measurement flights were embedded and the examined targets (Sect. 2.1). In Sect. 2.2, the participating instruments and collected data sets are summarized. Section 2.3 presents the flight strategy, which was used surveying the emission sources.

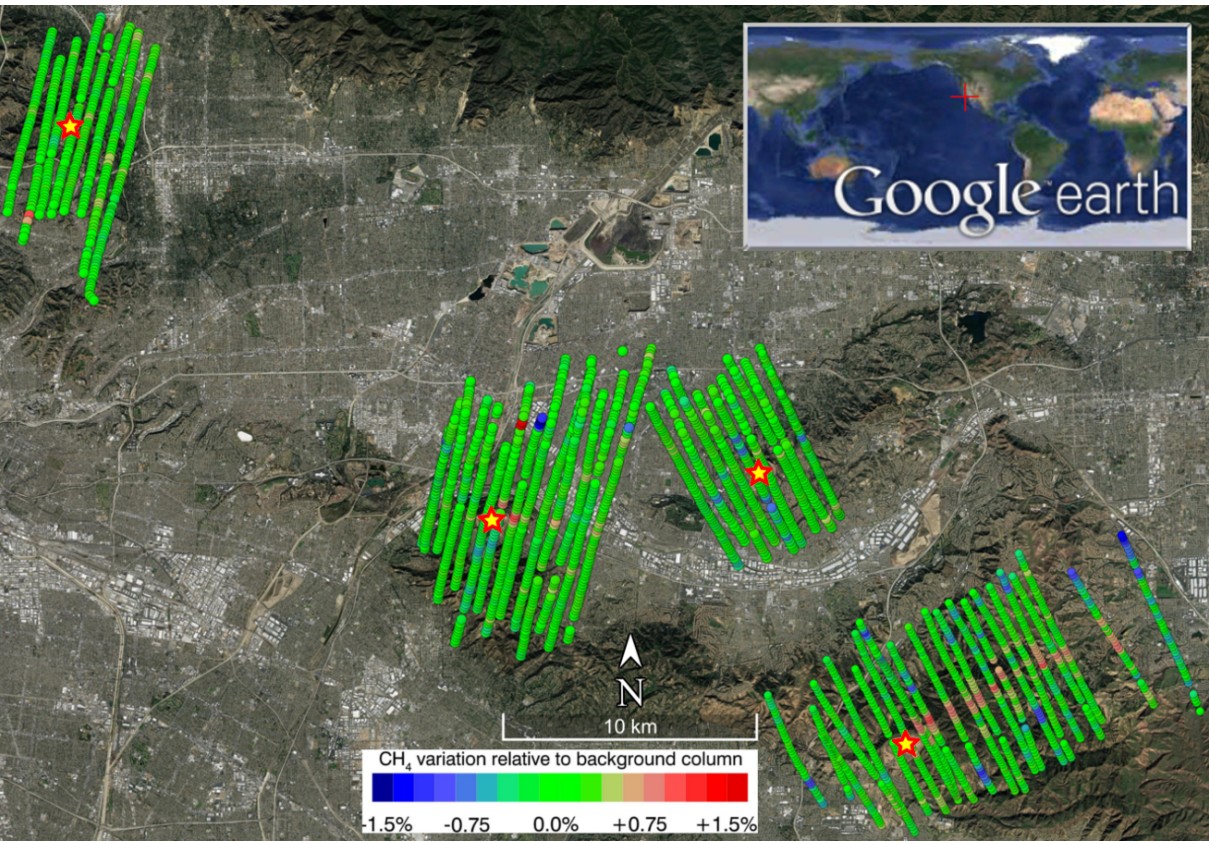

**Figure 1.** Shown are the MAMAP remote sensing survey flights over the four landfills (from left to right: the Scholl Canyon Landfill (SCL) on 27.08.2014, the Puente Hills Landfill (PHL) on 27.08.2014, the BKK Corporation Landfill (BKK) on 01.09.2014 and the Olinda Alpha Landfill (OAL) on 01.09.2014) situated in the Los Angeles Basin (red cross in the world map, top right). The locations of the landfills are marked by red/yellow stars. The MAMAP measurements are filtered by inclination to remove the turns and the colour code depicts $CH_4$ variations relative to the background (for details, see Sect. 3.1). Only the Olinda Alpha Landfill shows a clear plume in downwind direction. The wind direction was in general from south-west during the measurements. The map underneath is provided by Google Earth.

## 2.1 Campaign and target description

The measurement flights presented in this work were part of the $CO_2$ and Methane EXperiment (COMEX), which was conducted in the San Joaquin Valley (SJV) and greater Los Angeles in May/June and August/September 2014. COMEX was a collaborative effort between the National Aeronautics and Space Administration (NASA) and the European Space Agency (ESA), in support of the development of the two satellite concept missions HyspIRI (Lee et al., 2015) and CarbonSat (Bovensmann et al., 2010; Buchwitz et al., 2013; ESA, 2015). One focus of the campaign addressed the assessment of anthropogenic $CH_4$ emissions. In addition to measurements over and in the plumes of landfills, flights were conducted to determine emissions from oil fields (Thompson et al., 2015; Gerilowski et al., 2014), offshore seep fields and animal husbandry.

In total four different landfills were surveyed in the greater LA area: the Scholl Canyon Landfill (SCL), the Puente Hills Landfill (PHL), the BKK Corporation Landfill (BKK) and the Olinda Alpha Landfill (OAL). According to the US Environmental Protection Agency[1] (EPA), the yearly expected $CH_4$ emissions of these landfills were $5.0 \, \mathrm{kt} \, CH_4 \, \mathrm{yr}^{-1}$ (PHL), $5.9 \, \mathrm{kt} \, CH_4 \, \mathrm{yr}^{-1}$ (SCL), $11.5 \, \mathrm{kt} \, CH_4 \, \mathrm{yr}^{-1}$ (OAL) and $15.1 \, \mathrm{kt} \, CH_4 \, \mathrm{yr}^{-1}$ (BKK) in 2014 (further details on the reported emission rates can be found in Sect. S9 in the supplementary material). During remote sensing surveys only the Olinda Alpha Landfill continuously showed detectable and well-developed plume structures, which were well-suited for inversion of emission rates. The other landfills exhibited either much less pronounced measured enhancements (PHL) or no detectable enhancements in the remote sensing data at all (SCL and BKK, compare to Fig. 1). As this study investigates the use and ability of remote-sensing measurements in the SWIR region to determine emission rates, we focus on the data sets collected over the Olinda Alpha Landfill in the remaining manuscript. The OAL data sets were also the most comprehensive ones allowing for comparisons of emission rates on four different days. A discussion regarding the three other landfills is given in Sect. 4.7.

Concerning the Olinda Alpha Landfill, all measurements showed a pronounced $CH_4$ plume over the investigated time period. The landfill is located in Orange County, Los Angeles Basin, CA, USA (at $33.939°$ N, $117.836°$ W; Fig. 2). Measurements were acquired on four different days in the middle of the afternoon, during the last week of August and the first days of September 2014. On flight days, skies were clear and winds were from south-west to west at around 4 to $8 \, \mathrm{m \, s}^{-1}$.

The Olinda Alpha Landfill started operation in 1960 and is expected to close by 2030. It accepts a maximum of 8,000 $\mathrm{tons}$ of municipal solid waste daily and occupies an area of around $2.3 \, \mathrm{km}^2$, whereas $1.7 \, \mathrm{km}^2$ are used for waste disposal. Since 2012, a $32.5 \, \mathrm{MW}$ combined cycle power plant has been using the LFG to generate electricity for around 22,000 homes[2]. According to EPA, the estimated $CH_4$ amount released into the atmosphere was $11.5 \mathrm{kt} \, CH_4 \, \mathrm{yr}^{-1}$ in 2014 dropping from a peak value of $15.4 \, \mathrm{kt} \, CH_4 \, \mathrm{yr}^{-1}$ in 2011 to $14.3 \, \mathrm{kt} \, CH_4 \, \mathrm{yr}^{-1}$ in 2013.

## 2.2 Aircraft instrumentation and collected data sets

All instruments used for a quantitative analysis were flown aboard a DHC-6 Twin Otter (TO) aircraft operated by the Center for Interdisciplinary Remotely-Piloted Aircraft Studies (CIRPAS[3]). These comprise: the Methane Airborne MAPper (MAMAP), a Picarro CRDS greenhouse gas in-situ analyser, and the CIRPAS aircraft standard research instrumentation suite including different positioning and attitude, meteorological, aerosol, cloud and precipitation sensors.

The remote sensing instrument MAMAP (Gerilowski et al., 2011) was developed by the Institute of Environmental Physics (IUP), University of Bremen, in cooperation with the German Research Centre for Geoscience (GFZ) in Potsdam. It measures reflected and scattered solar radiation from the surface in the spectral region between 1.59 and $1.69 \, \mathrm{\mu m}$ at medium spectral resolution of around $0.9 \, \mathrm{nm}$ to retrieve total column concentration information of $CH_4$ and $CO_2$. In case of $CH_4$, the precision of the retrieved columns has been estimated to be better than $0.4 \, \%$ over land surfaces (Krings et al., 2013). For the current flights, a fibre coupled entrance telescope (connecting the telescope via a $5 \, \mathrm{m}$ glass fibre bundle to the spectrometer) was

---

[1] https://ghgdata.epa.gov/ghgp/main.do#/facility/, last access: 10.05.2017

[2] http://oclandfills.com/landfill/active/olindalandfill, last access: 21.06.2016

[3] http://www.cirpas.org/, last access: 17.10.2016

installed on a gyro stabilized platform (SOMAG, type: CSM-130[4]) to ensure nadir viewing geometry. The column information derived from MAMAP was used in combination with knowledge of the wind fields for the calculation of emission rates.

The Picarro fast greenhouse gas in-situ analyser (type: G-2301f) was provided by the NASA's Ames Research Center (ARC) and operated by IUP during the flights. The instrument uses the CRDS technique (Crosson, 2008) to measure $CH_4$, $CO_2$ and water vapour ($H_2O$) in-situ concentrations at flight altitude at a frequency of around $0.5\,Hz$. The flow rate of the installed external pump was around 165 standard cubic centimetres per minute (sccm) for altitudes between 600 and $1400\,m$ above sea level (m asl). In combination with a cavity volume of around $4.7\,cm^3$ at standard temperature and pressure (STP, $T = 0\,°C$, $p = 1013.15\,hPa$), this led to a flushing time and refilling time, respectively, of the cell of around $1.8\,s$. This value is close to the actual measurement frequency of the instrument. The air samples entered the aircraft through an atmospheric in-situ sampling boom and then were transported via a PTFE tubing system to the measurement cavity of the CRDS instrument. This process induced a time delay (in the following referred to as time lag) between the position where the air samples were acquired in the atmosphere and the time of measurement in the ring-down cavity of the instrument aboard the aircraft. This time lag was estimated from measurements in the laboratory to be around $21\,s$ with an associated uncertainty estimated to be $\pm 5\,s$. Dry gas mixing ratios of $CH_4$ and $CO_2$ were calculated by the software of the analyser via the synchronously measured water vapour (Rella et al., 2013, and references therein). The uncertainties of the dry gas mixing ratios of $CH_4$ and $CO_2$ have been estimated to be $2.3\,ppb$ and $0.15\,ppm$, respectively, from laboratory experiments. The dry gas mixing ratios have also been assessed against known National Oceanic and Atmospheric Administrations (NOAA) standards. The resulting calibration factors for $CH_4$ (1.002275041) and $CO_2$ (1.004664623) were applied to correct the dry gas mixing ratios in advance to further analysis. These measurements were used for an independent in-situ based emission estimate. This enabled a comparison to be made with the emission estimated by the MAMAP remote sensing data.

The CIRPAS aircraft standard research instrumentation suite delivers auxiliary data. These comprise, for example, 3D position information (attitude, heading), wind information (speed and direction derived from a 5-hole turbulence probe), and information for the characterization of the atmosphere (e.g., potential temperature, aerosol load, ambient temperature, pressure) at a frequency of 10 or $1\,Hz$ depending on the measured parameter.

On one flight day, the CIRPAS TO was accompanied by a second Twin Otter aircraft flying the Airborne Visible / Infrared Imaging Spectrometer - Next Generation (AVIRIS-NG; Green et al., 1998; Hamlin et al., 2011) operated by the Jet Propulsion Laboratory (JPL). AVIRIS-NG also measures backscattered solar radiation from the surface to infer column information on $CH_4$. In contrast to MAMAP, AVIRIS-NG is an imaging instrument with a high spatial sampling but relatively low spectral resolution of $5\,nm$ and a wide spectral range from 0.38 to $2.51\,µm$. Typical $CH_4$ retrievals use the spectral region from 2.1 to $2.4\,µm$. In this study, we used the AVIRIS-NG instrument's imaging capabilities to identify potential source position(s) of $CH_4$ emitted by the landfill, which was not possible with the non-imaging MAMAP instrument (Thompson et al., 2015).

---

[4]http://www.somag-ag.de/csm-130/, last access: 02.08.2016

## 2.3 Flight strategy

To achieve the goal of estimating the emission rate of an areal source like a landfill (here: around $1.7\,km^2$) using combined aircraft remote sensing and in-situ observations by the MAMAP and Picarro instruments, an appropriate flight pattern needed to be flown by the aircraft. The measurements during the flights were divided into two parts: 1) remote sensing measurements of the $CH_4$ plume from above the atmospheric boundary layer covering the entire area by a dense pattern and 2) in-situ measurements intersecting the entire plume within the atmospheric boundary layer.

The remote sensing and in-situ flight patterns implemented during the campaign were developed and optimized during pre-flight planning on the basis of the above requirements as well as taking into account the weather forecast, restrictions due to Air Traffic Control (ATC), and available flight time. Each acquisition started with remote sensing measurements and was followed by in-situ measurements.

Performing the remote sensing measurements first had a significant advantage as a $CH_4$ real-time retrieval utilizing the MAMAP data had been implemented for the COMEX campaign. In case the real wind direction deviated from the forecast, this approach allowed the operator to dynamically adjust the flight pattern accordingly to match the plume location obtained from the remote sensing total column information. The latter was dynamically superimposed on Google Earth map data.

The remote sensing tracks were typically flown above the atmospheric boundary layer in a dense pattern perpendicular to the wind direction covering the entire measurement area. In-situ $CH_4$ and $CO_2$ data were also acquired during these measurements providing information on $CH_4$ and $CO_2$ concentration distributions in the measurement area above the boundary layer.

To ensure a good coverage of the vertical extent of the plume during the second part of the flight focusing on in-situ measurements, the aircraft typically flew at a fixed distance from the source for several plume transects perpendicular to the prevailing wind direction at different altitudes trying to best cover the entire boundary layer. The number of legs for such a "wall" of measurements varied depending on the available flight time, between 3 and 6. Additionally, depending on available flight time such a wall was typically flown upwind and downwind characterizing the inflow and outflow to the area. On one day, one additional downwind wall of measurements was located at a distance further away from the source to better characterize occurrent errors on the estimated fluxes. The maximum altitude extent of the plume was generally well documented, as on all four flight days, there was at least one leg, which shows no plume structures or signals at higher altitudes and therefore confines the upper limit of the plume. Due to ATC restrictions over congested areas like the Los Angeles Basin, flying below $1000\,ft$ above ground level (ft agl, equals around $300\,m$ agl) was not permitted. Therefore, the lowest measured track was typically extrapolated down to the surface following the terrain (more details are given in Sect. 4.2). Altitude changes were made not faster than 150 metres per minute to minimise the effect of pressure changes on the in-situ sampling. This rate of change maintained the sampling cavity conditions well within acceptable tolerances: i.e. Cavity pressure within $140.0\pm0.04\,Torr$ and cavity temperature within $45.0\pm0.002\,°C$ (deviations are given in $\pm1$-$\sigma$). Figure 2 shows the approximate position of the three upwind (dashed lines) and five downwind (solid lines) walls flown on all four days.

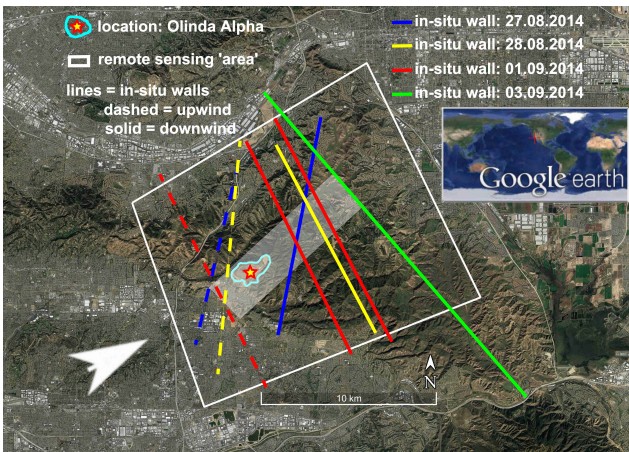

**Figure 2.** The Olinda Alpha Landfill is located at the position of the red/yellow star encircled by the cyan solid line. Additionally, the approximate positions of the flown in-situ upwind (dashed lines) and downwind (solid lines) walls (see Sect. 2.3 for details) and the area which were surveyed by the MAMAP remote sensing instrument (white box) are shown. The colours of the in-situ walls represent the different flight days at the Olinda Alpha Landfill: Blue: 27.08.2014. Yellow: 28.08.2014. Red: 01.09.2014. Green: 03.09.2014. The white arrow indicates the approximate prevailing wind direction for measurement flights over the Olinda Alpha Landfill .The transparent white area aligned in the approximate wind direction shows one example flight track of the AVIRIS-NG imaging instrument abroad the second Twin Otter aircraft for illustration purposes. The map underneath is provided by Google Earth.

The flight pattern performed by the second aircraft with the imaging AVIRIS-NG instrument aboard was different. Due to its relatively wide swath, it needed only one flight line to cover the entire landfill. Measurements were acquired while flying well above the boundary layer approximately parallel to the prevailing wind direction.

## 3 Retrieval algorithms and calculation of emission rates

This section presents the steps necessary to obtain the emission rates from the measurements collected by the different instruments. Section 3.1 describes the MAMAP retrieval algorithm and the assessment of the emission rate estimates including discussions of associated uncertainties (Sect. 3.1.1) and possible dependencies of the retrieved columns on the detector filling (Sect. 3.1.2). Section 3.2 explains how the in-situ data collected by the Picarro greenhouse gas in-situ analyser was used to determine emission rate estimates and related uncertainties (Sect. 3.2.1). In addition, the retrieval of the $CH_4$ anomaly maps from the AVIRIS-NG imaging data is described in Sect. 3.3.

### 3.1 MAMAP retrieval algorithm and emission rate estimates

In order to retrieve the column amounts of $CH_4$ and $CO_2$ from the measured spectra, we used the Weighting Function Modified Differential Optical Absorption Spectroscopy (WFM-DOAS) method (for details, see Buchwitz et al., 2000; Krings et al., 2011, 2013). In general, the algorithm minimizes the differences between the logarithm of the measured spectra and a spectrum

computed by a Radiative Transfer Model (RTM), which describes the general or mean state of the atmosphere during the flight. The differences between the modelled and measured spectra are minimized by varying selected parameters or fit factors on, e.g., the methane profile and atmospheric parameters. The $CH_4$ in the plume from landfill emissions is then seen to be enhanced relative to the surrounding air.

The WFM-DOAS algorithm has successfully been applied to aircraft MAMAP measurements and used to investigate the emissions of point sources at known locations having flue gas or ventilation stacks / chimneys with diameters of below $50\,m$, such as those from $CO_2$ emitting power plants (Krings et al., 2011, 2016) and $CH_4$ emitting ventilation shafts of coal mines (Krings et al., 2013). In contrast to the previous studies, we have applied the approach to an areal source i.e. a landfill (with a size of around $1.7\,km^2$), where the exact locations of the emission(s) are not known but limited by the approximate area of the

landfill. As a result of the larger area of this source and the resulting wider plume, the expected column enhancements within the plume are typically lower in comparison to enhancements produced by point sources with diameters smaller than $50\,m$ having the same source magnitude for similar atmospheric conditions.

For each flight a dedicated set of RTM computations were calculated to account for the varying atmospheric conditions on the different days. Additionally, a change in the solar zenith angle (SZA) and surface elevation along the flight track were

taken into account by performing RTM simulations to generate a 2-dimensional look-up table, which was then used in the retrieval. The surface elevation is based on data from the Shuttle Radar Topography Mission (SRTM) digital elevation model version $2.1$[5] which has a spatial resolution of one arc second (around $30\,m$ at the equator) in the U.S.A. Additionally, the remaining parameters flight altitude, surface albedo and atmospheric background profiles were also adapted to the current flight conditions:

– The surface was assumed to have a Lambertian reflectance and for the spectral band of $CH_4$ and $CO_2$ to have no spectral dependency. The surface spectral reflectance or surface albedo values used were taken from Chen et al. (2006). They used clear-sky radiances measured by the Moderate Resolution Imaging Spectrometer (MODIS) onboard the Terra satellite and the Visible Infrared Scanner (VIRS) onboard the Tropical Rainfall Measuring Mission (TRMM) spacecraft to retrieve the surface albedo in different spectral channels for different surface type categories defined by the International

Geosphere Biosphere Programme (IGBP). As the MAMAP instrument operates in the SWIR region around $1.65\,\mu m$, we used the surface albedo derived from the $1.6\,\mu m$ channel of MODIS and VIRS. Assuming that the surface type at and around the landfill can be described as a composite of approximately $50\,\%$ 'urban' and $50\,\%$ 'open shrubland' (corresponding to a retrieved surface albedo of around 0.22 and 0.40 (Chen et al., 2006, their Table 1), respectively), this yields a mean surface albedo of 0.31.

– For the background profiles of $CH_4$ and $CO_2$ (compare to Fig. S19 in the supplementary material), which describe the mean background concentrations of these gases in the measurement area (and are not influenced by, for example, the landfill emissions) the vertical profiles from the U.S. standard atmosphere were used and adapted to current concentrations by using data collected by the Picarro greenhouse gas in-situ analyser. The profiles of $CH_4$ and $CO_2$ in the lower

_______________
[5]http://dds.cr.usgs.gov/srtm/version2_1/, last access: 15.06.2016

part of the troposphere were replaced by a polynomial fitted to the measured profile corresponding to in-situ measurements collected at the respective site. In-situ measurements gathered at remote sensing altitude were assumed to belong to the free troposphere and, thus, were used to scale the entire upper part of the U.S. standard profiles.

The HITRAN 2012 spectroscopic database for line parameters (Rothman et al., 2013) and a standard OPAC (Optical Properties of Aerosols and Clouds) urban aerosol scenario (Krings et al., 2011, 2013, 2016) were used in the RTM calculations, as the landfill is located within Los Angeles Basin.

The column-averaged dry air mole fractions $XCH_4$, which were used in the estimation of the $CH_4$ emission rate, were retrieved utilising the $XCH_4(CO_2)$ proxy method. This assumes a spatially and temporally constant $CO_2$ background concentration in the measurement area during the time remote sensing measurements are taken. In contrast to Krings et al. (2013), where coal mine ventilation shafts emitted only $CH_4$ and no significant amounts of $CO_2$, this assumption is violated for the Olinda Alpha Landfill. For landfills, it is expected that the co-emitted $CO_2$ may have an influence on the obtained $XCH_4(CO_2)$ (or short $XCH_4$) columns when this proxy method is used. The impact is further investigated in Sect. 3.1.1.

The procedure to estimate the $CH_4$ emissions from the retrieved MAMAP $XCH_4$ data comprised the following steps. The data was first filtered by a signal filter to remove spectra with very low detector filling (less than 3000 counts) or spectra in saturation (as in Krings et al., 2011). Additionally, an inclination filter of $\pm 5°$ was applied to eliminate measurements during aircraft turns or insufficient gyro stabilization by the CSM-130. Furthermore, the data obtained for each flight track was normalised by data obtained at its edges / flanks outside the plume (similar to Krings et al., 2016). This step was necessary to remove a possible constant offset from the data (see also Krings et al., 2011) and to account for potential horizontal $CH_4$ or $CO_2$ concentration gradients.

Based on these measured column-averaged dry air mole fractions, $XCH_4$ (or $CH_4$ variations relative to the background column), a flux corresponding to each track was estimated by applying a mass balance approach (similar to that used in Krings et al., 2011, 2013, 2016):

$$F_{RS} = f_{RS} \cdot \frac{1}{n} \sum_i^n u_{perp,i} \sum_j^{k_i} V_{i,j} \cdot \Delta x_{i,j} \tag{1}$$

where $n$ is the total number of flight tracks downwind of the landfill flown on a certain day, $k_i$ is number of measurements of a certain flight track $i$, $V$ is the retrieved $CH_4$ variation relative to the background column in $molec\,m^{-2}$ of measurement j for track number i, $\Delta x_{i,j}$ is the length segment in m of a certain measurement j of track number i, $u_{perp,i}$ is the wind speed component perpendicular to the flight track $i$ in $m\,s^{-1}$, $f_{RS}$ is a conversion factor including the mass per $CH_4$ molecule and the time conversion from s to yr ($8.398 \cdot 10^{-25}\,kt\,s\,molec^{-1}$) in order to calculate the emission rate $F_{RS}$ in $kt\,CH_4\,yr^{-1}$ based on the MAMAP remote sensing measurements. The emission rate is given in $kt\,CH_4\,yr^{-1}$ but is strictly speaking only valid for the time of the overflight.

As in previous studies, the required wind direction was directly estimated from the measurements (observed plumes) themselves. The wind speed was provided by the 5-hole turbulence probe of the CIRPAS instrumentation, whereas only wind

measurements collected in the area of the plume are used. Further details of the definition of the plume area for the wind estimates are given in Sect. 4.2.

### 3.1.1 Uncertainties of estimated MAMAP remote sensing emission rates

The largest errors or uncertainties for the remote sensing based emission estimates originate from uncertainties of the wind parameters used (wind speed and direction), the chosen concentration background normalisation area, the track-to-track variability, the influence of $CO_2$ variations in terms of the applied $XCH_4(CO_2)$ proxy method and the used surface albedo in the RTM simulations. In the following, the methodology of how these uncertainties were quantified is discussed. The resulting uncertainties are then given in Sect. 4.1 together with the estimated emission rate estimates for the single days.

A wind speed error linearly propagates into the emission estimate (compare to Eq. 1). As the in-situ measurements of the 5-hole turbulence probe were utilized for the wind speed estimates, the accuracy of the probe was used as a first order approximation for an uncertainty estimate. The uncertainty of the turbulence probe wind speed data has been estimated to be $0.5\,\mathrm{m\,s^{-1}}$.

The wind direction enters the flux estimate via a cosine term by modifying the used perpendicular wind speed to each flight track. An error on the wind direction of $\pm 10°$ was assumed for the case when wind direction is derived from the measurements themselves.

The lateral positions used for the background normalisation area may also have an influence on the result. In order to test their impact on the final emission estimate, the limits were shifted towards or away from the center line by a certain distance. For this type of test, one needs to keep in mind that if the limits are too close to the plume, part of the plume signal may enter the area used for the background normalization leading to an underestimate of the emission. On the other hand, if the limits are set too far away, there might be not sufficient measurements left to calculate a reliable concentration background. Thus, the limits were varied by $\pm 250$ and $\pm 500\,\mathrm{m}$ and, additionally, the defined plume area was shifted as a whole by 250 and 500 m to the right and left with respect to the center line.

Additionally, we computed the statistical error contribution. This error source is referred to as track-to-track variability in the following. Based on the used downwind tracks, a standard deviation $\sigma$ and from that the uncertainty of the mean, was calculated (for further details, see also Farrance and Frenkel, 2012).

For the remote sensing emission rate estimate, the $XCH_4(CO_2)$ columns, determined using the proxy method, were used. The proxy method assumes that $CO_2$ is equally distributed and did not change in the measured area during the flight. In general, any $CO_2$ enhancement would lead to a decline in the derived $XCH_4(CO_2)$. The influence of such a $CO_2$ anomaly on the emission rate estimate depends on its location. On the one hand, the $CO_2$ enhancements can be co-located to the $CH_4$ landfill plume for the case when the $CO_2$ is co-emitted. This will lead to an underestimation of the emission rate. On the other hand, if the $CO_2$ originates from outside the measurement area, the enhancement is not co-located to the $CH_4$ plume. This results in an under- or overestimation of the emission rate depending on the location and distribution of the $CO_2$ variations. To estimate the influence of a variable $CO_2$ concentration in the measurement area on the remote sensing emission rate estimates, integrated in-situ columns (IISCs, compare to Fig. 8) were derived for the measured in-situ walls (compare to Fig. 5, b). The in-situ $CH_4$

and $CO_2$ measurements were vertically integrated from the surface to the highest altitude of the in-situ wall. Subsequently, the two obtained IISCs for $CH_4$ and $CO_2$ were similarly treated as they would be in the MAMAP proxy approach. First, the $CH_4$ column was divided by the $CO_2$ column and then the track was background normalized by its edges. This results on the one hand in an $\mathrm{IISC_{CH_4}}$ from the $CH_4$ enhancement only, which is not influenced by $CO_2$ variations, and on the other in an $\mathrm{IISC_{CH_4/CO_2}}$ which considers $CO_2$ variations. To quantitatively estimate the influence of this offset on the final emission rate estimate, the emission through each in-situ based cross-section $\mathrm{IISC_{CH_4}}$ and $\mathrm{IISC_{CH_4/CO_2}}$ was calculated by using Eq. 1. The column enhancement $V$ and the length segment $\Delta x$ are given by Fig. 8, whereby the remaining parameters, especially the perpendicular wind speed, cancel out, because we are only investigating in the relative difference.

An error of a wrongly assumed surface albedo in the simulated RTM, which is used during the fit procedure, is expected to have only a small influence on the estimated emission rate because it is captured by a low order polynomial, which is used during the retrieval process (also compare to Krings et al., 2011). To investigate the influence of a wrongly assumed surface albedo, emission rates were also determined based on RTM simulations using albedos of 0.22 and 0.40 representing the pure 'urban' and 'open shrubland' scenarios, respectively.

The total uncertainties were calculated by root-sum-squaring the single uncertainties for each day with the underlying assumption that the error sources were not correlated. The resulting total uncertainties including the uncertainties in wind information, normalization area, track-to-track variability, $CO_2$ variations, and surface albedo of the remote sensing measurements are stated in Sect. 4.1.

### 3.1.2 Non-linearity and associated negative $XCH_4$ anomalies

When investigating the retrieved normalised column-averaged dry air mole of $CH_4$ from the MAMAP remote sensing measurements on the 01.09.2014 (Fig. 6, a), they also show, besides a clear plume structure downwind of the landfill, some blue spots. First investigations have revealed some column dependencies on the detector filling. The scatter plot in Fig. 3 shows the ratio of the retrieved $CH_4$ and $CO_2$ profile scaling factors as a function of detector filling. It (black diamonds) clearly shows a decrease in the ratio for lower signals and also a less pronounced decrease for higher detector fillings. The cause of this dependency is still under investigation. The effect is most pronounced on the 01.09.2014 flight having the most measurements at lower detector fillings (e.g., $32\,\%$ below $13000\ \mathrm{counts}$) with respect to the three other days ($5\,\%$ on 27.08.2014, $12\,\%$ on 28.08.2014 and $2\,\%$ of the measurements on 27.08.2014). Therefore, the effect was further investigated exemplarily for the 01.09.2014.

In order to test the assumption that the negative $XCH_4(CO_2)$ anomalies originate from this signal dependency on the 01.09.2014, a $3^{rd}$ order polynomial (Fig. 3, red solid line) was fitted to the scattered data and subsequently used for correction. The new data set exhibits nearly no dependency on the detector filling (Fig. 3, green diamonds). Furthermore, the blue spots in Fig. 6 (b) are reduced compared to Fig. 6 (a). The $1\text{-}\sigma$ track-to-track variability has also been reduced by $26\,\%$.

It is expected that this effect was less relevant for measurements from previous campaigns because the measured radiance signals and column enhancements were significantly higher than in this study. The mean estimated emission rate has further-

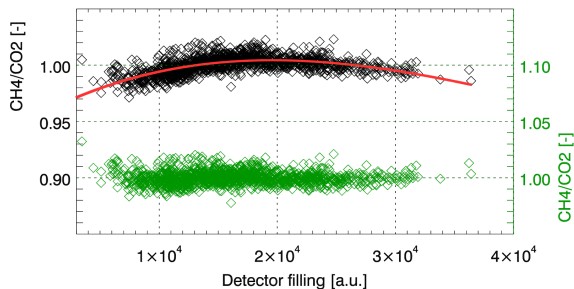

**Figure 3.** Scatter plot of the ratio of the retrieved $CH_4$ and $CO_2$ column over the maximum detector filling on the 01.09.2014. Black diamonds: Non-corrected data, left scale. Red solid line: Fitted $3^{rd}$ order polynomial. Green diamonds: Corrected data, right scale.

more changed by less than $2\,\%$ for the investigated Olinda Alpha Landfill measurements on the 01.09.2014 due to this effect and can therefore be neglected.

## 3.2 In-situ emission rate estimates by Picarro data

Fluxes from the Picarro data were estimated separately for each downwind wall by the procedure described below. An in-situ wall of measurements comprised several flight legs flown at different altitudes. Usually these flight legs were not aligned perfectly parallel to each other and separated by around $150\,m$ in altitude. For interpretation and estimation of reliable emission rates, the in-situ measurements were projected on a well-defined plane and perpendicular surface and the gaps between different tracks were filled by inter- and extrapolating, respectively, the measurements to a regular 2D grid on that plane.

Before the measurements from the flight legs of each wall were projected onto the plane surface, which is called in-situ wall in the following, they were first corrected for the time lag of $21\,s$ resulting from the tubing system (Sect. 2.2). The approximate positions of those in-situ walls are drawn in Fig. 2. The projection of the $CH_4$ measurements is shown in Fig. 5 (a) for the first downwind wall on the 01.09.2014. Figure 5 further comprises (b) the interpolated $CH_4$ mixing ratios, (c) the background $CH_4$ mixing ratios and (d) the enhanced $CH_4$ mixing ratios attributed to the plume of the landfill resulting from the next processing steps described in the following.

For the inter- and extrapolation, the statistical Kriging method (Krige, 1951) was chosen. A similar approach was also used in, e.g., Mays et al. (2009), Cambaliza et al. (2014) and Lavoie et al. (2015), to determine the outflow of cities and emissions of landfills. It is used to estimate values at locations, where no sample was measured (in our case, mostly between the projected flight legs), with the aid of statistical methods. This method is described by the three parameters nugget, sill and range, which describe the statistics of the data set. The nugget stands for the small scale variability, the sill is the variance and the range gives the distance at which the samples are not correlated any more.

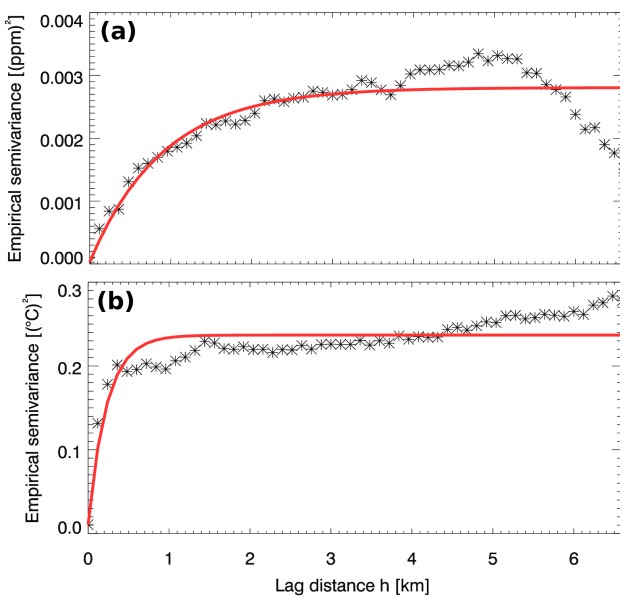

**Figure 4.** Example experimental semivariograms of (a) the in-situ dry gas mixing ratio of $CH_4$ and (b) the ambient temperature for the second downwind wall of the Olinda Alpha Landfill measurements on 01.09.2014. The black crosses depict the values of the empirical semivariance at certain lag distances $h_j$ and the solid red line is the fitted exponential function. The fitted parameters of the exponential model are: Range: (a) 2.7 km, (b) 0.7 km; partial sill: (a) $2.8 \cdot 10^{-3}$ $ppm^2$, (b) $2.3 \cdot 10^{-1}$ $°C^2$; nugget: (a) $3.0 \cdot 10^{-5}$ $ppm^2$, (b) $1.1 \cdot 10^{-2}$ $°C^2$.

All three parameters can be inferred from an experimental semivariogram (Fig. 4) calculated by the following equation (e.g., after Isaaks and Srivastave, 1989; Cressie, 1993; Caers, 2011):

$$y(h_j) = \frac{1}{2N(h_j)} \sum_{N(h_j)} \left[ V(s_i) - V(s_i + h_j) \right]^2 \tag{2}$$

where $h_j$'s are equidistant lag distances (e.g., $\ldots 360\,\text{m}$, $480\,\text{m}$, $600\,\text{m}$, $\ldots$) which are separated by a constant lag separation
5  distance or bin width $h_\text{sep}$ (e.g., $120\,\text{m}$). The lag distance $h_j$ describes the distance of the position between two measurements for which the semivariogram value $y(h_j)$ is calculated (Fig. 4, black crosses), whereby $N(h_j)$ is the number of data pairs for the respective lag distance $h_j$ and the sum denotes the summation over all data pairs $i$ which are separated by a certain lag distance $h_j$. $V(s_i)$ and $V(s_i + h_j)$ are the parameter values at the positions $(s_i)$ and $(s_i + h_j)$ separated by one specific lag distance $h_j$. For an irregularly spaced sample either a lag tolerance is introduced to consider also measurements, which are
10  located in the approximate position of $h_j$, or the bin width itself is used meaning all measurements between $h_j$ and $h_{j+1}$ are considered.

The experimental semivariogram was calculated for each wall and for each parameter by an IDL routine written by James McCreight from the University of Colorado in 2008[6] after the projected measurements of the corresponding parameter were detrended. In general, the semivariogram describes the correlation between different points at different distances.

---

[6]https://github.com/mccreigh/idl_variogram, last access: 06.07.2016

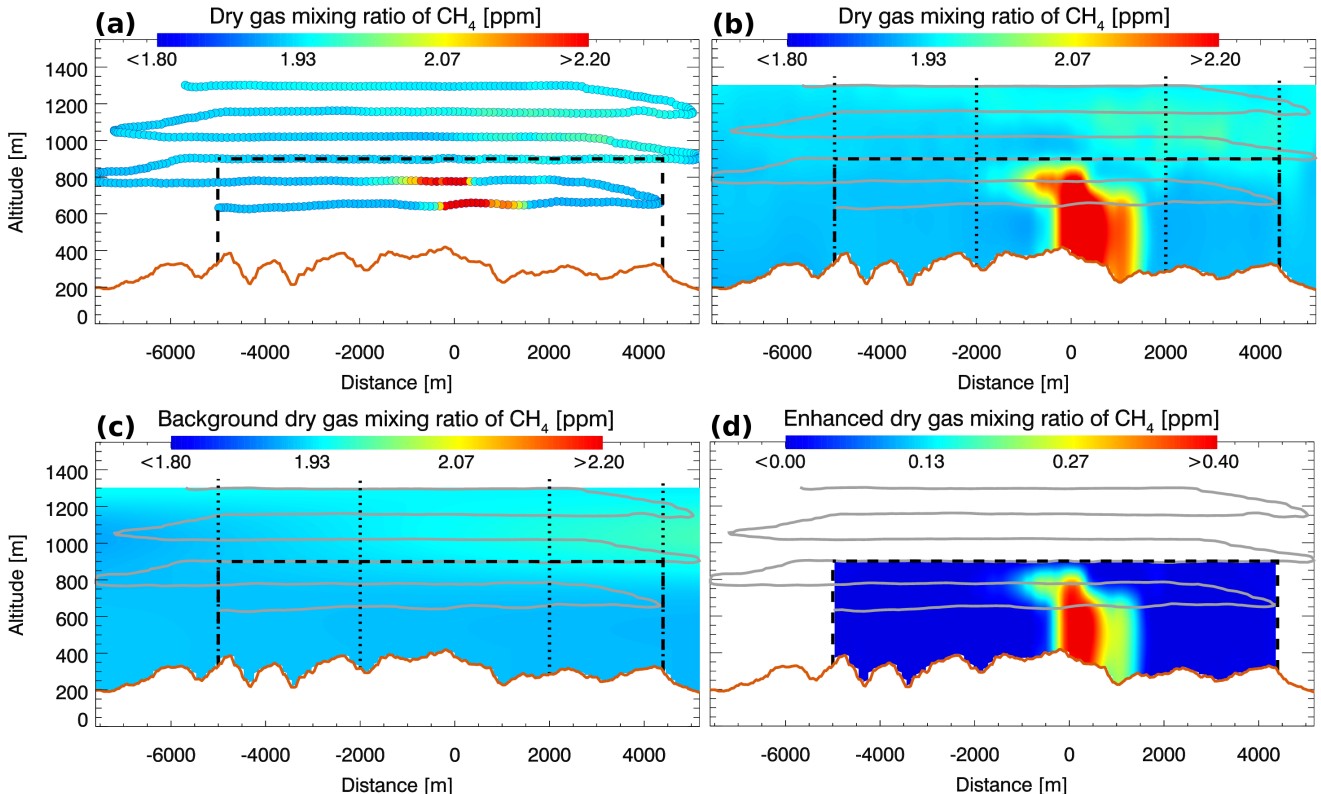

**Figure 5.** Example dry gas mixing ratios of $CH_4$ for the first downwind wall measured on the 01.09.2014 (middle in-situ wall in Fig. 6, solid red line). (a) Projected and time lag corrected mixing ratios acquired along the flight track onto the wall. (b) Kriged mixing ratios based on the measurements in (a) and an additionally added pseudo-track at the surface (not shown, see Sect. 4.2 for details). (c) Derived background mixing ratios from (b). (d) Derived $CH_4$ enhancement (kriged mixing ratios in (b) minus background mixing ratios in (c)). X-axis gives the distance from the approximate plume centre in m and y-axis gives the altitude in m above sea level (m asl). Solid orange lines depict the surface elevation (based on SRTM) and solid grey lines the projected flight track. Vertical dotted black lines show horizontal limits, which were used to define the background area (here: from - 5.0 to - 2.0 km and from +2.0 to +4.4 km). The area, which was used in the mass balance approach for estimating the emission rate, is enclosed by the dashed black lines.

To this experimental semivariogram, a commonly used exponential model function (e.g., after Isaaks and Srivastave, 1989) was fitted (Fig. 4, red solid line) which yields the necessary parameters range, nugget and (partial) sill:

$$model = nugget + partial\ sill \cdot \left[ 1 - e^{-\frac{3h}{range}} \right] \qquad (3)$$

In this model, the value of the nugget is given by the value of the experimental semivariogram at the origin, the value of the sill corresponds to the sum of the nugget and the fitted parameter partial sill, and the range is defined as the lag distance h at which 95 % of the sill is achieved (Journel and Huijbregts, 1978).

The parameters from the exponential model were used to estimate the value $V(s_0)$ of the paramters, e.g., the dry gas mixing ratio of $CH_4$, at a position $s_0$ where no measurement had been acquired based on the measured surrounding values $V(s_i)$ at the positions $s_i$:

$$V(s_0) = \sum_{i=1}^{n} w_i \cdot V(s_i) \tag{4}$$

The influence of measured values $V(s_i)$ on the result is described by the respective weights $w_i$, whereas $n$ is the total number of measurements. The weights are determined on basis of the above calculated parameters for the exponential model and the distances between the measured values and the unknown value, respectively (for further details, see Isaaks and Srivastave, 1989). Equation 4 was evaluated for each grid point on the plane surface.

For computation, the Kriging procedure 'Krig_2D' from IDL 8.2.3 was used[7]. An example of such a kriged in-situ wall is
shown in Fig. 5 (b) for $CH_4$ measurements of the first downwind wall on the 01.09.2014.

Subsequently, the mass transport of $CH_4$ through each wall was estimated by a mass balance approach:

$$F_{IS} = \Delta z \cdot \Delta x \cdot f_{IS} \sum_i (c_i - c_{0,i}) \cdot \frac{p_i}{T_i \cdot k_B} \cdot u_{eff,i} \tag{5}$$

where i is the index representing the $i^{th}$ grid box, $c$ is the measured $CH_4$ concentration in $\mu mol\,mol^{-1}$ or ppm, $c_0$ is the $CH_4$ background concentration in $\mu mol\,mol^{-1}$ or ppm, $p$ is the pressure in Pa, $T$ the ambient temperature in K, $k_B$ the Boltzmann
constant, $\Delta z$ and $\Delta x$ are the vertical and horizontal extents of the grid boxes in m, respectively, $f_{IS}$ is a conversion factor having the same value and units as $f_{RS}$ in Eq. 1 in order to retrieve the emission rate $F_{IS}$ in $kt\,CH_4\,yr^{-1}$, and $u_{eff}$ is the effective wind speed in $m\,s^{-1}$. The effective wind speed accounts for the wind speed normal to the plane surface and a geometry factor which considers the orientation of the wall relative to the orientation and flight direction of the aircraft, respectively, while a single measurement is recorded, and the wind direction. If the fitted wall is parallel to the measurement or perpendicular to the wind
direction, the geometry factors becomes 1. The concentration $c$, the temperature $T$ and the effective wind speed $u_{eff}$ are based on Kriging, whereas the pressure $p = p(z)$ only depends on the altitude of the grid box $i$. The functional dependency $p(z)$ has been determined beforehand by fitting a linear function to the projected pressure measurements.

As indicated by Eq. 5, only the $CH_4$ enhancement above the background is needed. In order to separate the plume signal from the background, the plane surface of the $CH_4$ measurements was segmented into a plume area and a background area (Fig.
5, b). For each altitude level, a linear function was fitted to the $CH_4$ measurements in the background area by a least-squares approach. This yields a 2D-distribution of the $CH_4$ background for the specific in-situ wall (Fig. 5, c). Subtracting the achieved $CH_4$ background from the plane surface of the $CH_4$ measurements results in the pure $CH_4$ signal (Fig. 5, d) originating from the source under consideration. This method accounts for possible concentration gradients in the $CH_4$ background in the horizontal and vertical direction.

---

[7]http://www.harrisgeospatial.com/docs/krig2d.html, last access: 04.03.2016

### 3.2.1 Uncertainties of estimated Picarro in-situ emission rates

For the error budget of the in-situ based emission rates, two groups of error sources were identified: a) measurement related uncertainties and b) method related uncertainties. In the following, the main error sources are shortly discussed. A summary of all resulting errors for the different downwind walls is given in Table 2.

One main contributor to group a) is the wind information, which enters Eq. 5. It is based on measurements taken by the 5-hole turbulence probe of the CIRPAS instrumentation. Any error in the wind speed linearly propagates to the emission estimate. In a first order approximation, the accuracy of $0.5\,\mathrm{m\,s^{-1}}$ of the turbulence probe was related to the averaged absolute wind speed of a downwind wall for estimating its influence on the estimated emission rate.

Another important error originates from the lack of measurements down to the surface. As baseline, it was assumed that the plume had been well-mixed in the lower part of the atmospheric boundary layer. On the one hand, $CH_4$ concentration might increase towards the surface because landfills are surface sources (Gordon et al., 2015). On the other hand, the in-situ walls were acquired some kilometres downwind of the landfill so that it is expected that some vertical mixing had occurred suppressing very high accumulations of $CH_4$ at the surface. To quantify these effects, it was assumed in a first order approximation that the pseudo-surface track used for extrapolation contains 50 or 150 % of the $CH_4$ enhancements with respect to the lowest observed flight track.

A third error source originates from the time lag, which was around $21\,\mathrm{s}$. The estimated uncertainty of the time lag was $5\,\mathrm{s}$. In order to assess the sensitivity of final emissions to a variation of the time lag, fluxes were estimated with time lags varying between 16 and $26\,\mathrm{s}$.

Group b) consists of errors which originate e.g. from the chosen interpolation technique "Kriging" and how these data were used in the mass balance approach.

As discussed in the previous section, the Kriging method requires the three parameters nugget, (partial) sill and range, which were derived beforehand by fitting an exponential function to the experimental semivariogram for each quantity used in the mass balance approach. To quantify the influence of the Kriging parameters on the estimated emission and how sensitive it responds, the range was varied by a factor of 4 (i.e., -75 % and +300 %). Additionally, six configurations for the parameters nugget and partial sill (bearing in mind that the sill is the sum of partial sill and nugget) were investigated. On the one hand, the nugget was set to zero so that the partial sill equalled the sill and on the other hand, the nugget was increased to half of the sill and the partial sill was decreased to half of the sill. This was done for three different sills: the standard derived sill, two times the standard derived sill and half the standard derived sill. Furthermore, the effect of a varying lag separation distance, which also slightly influences the fitted parameters, is covered.

A further error source originates from the limits for the background area. To test its sensitivity, the limits were varied till their size had only 50 % of the original size.

The above mentioned error sources were combined for calculating a total uncertainty of the estimated emission rate for each downwind wall. For that, the errors were assumed to be independent and root-sum squared. The uncertainties for the four flight days are listed in Table 2.

### 3.3 Retrieval of CH$_4$ anomaly maps by AVIRIS-NG data

AVIRIS-NG methane retrievals use a matched filter approach previously demonstrated in campaigns at Kern River (Thompson et al., 2015), Four Corners (Frankenberg et al., 2016), and Aliso Canyon (Thompson et al., 2016). We treat AVIRIS-NG spectra $x$ as Independent Identically Distributed (IID) instantiations of a multivariate Gaussian distribution with mean $\mu$ and
5    covariance matrix $\Sigma$, written $x \sim \mathcal{N}(\mu, \Sigma)$. To account for the independent noise properties of each detector element, we model the spectra from each pushbroom element separately. This produces a slightly different distribution for every cross-track position. The covariance matrices are regularized to ensure accuracy and numerical stability for the limited number of samples. For each new spectrum, the matched filter estimates the magnitude $\alpha$ of a linear perturbation of this Gaussian distribution in the direction of the target signal. The estimate $\hat{\alpha}(x)$ is written:

$$10 \quad \hat{\alpha}(x) = \frac{(x - \mu)^T \Sigma^{-1} t}{t^T \Sigma^{-1} t} \tag{6}$$

Here the target is the radiance Jacobian with respect to a change in CH$_4$ absorption above background. The magnitude of the resulting estimate indicates the enhancement of CH$_4$ absorption above the local background in units of ppm $\times$ meters. After detection, the resulting maps were georectified to permit direct comparison with MAMAP retrievals using synchronized IMU/GPS data and a local digital elevation model.

## 4    Results and discussion

### 4.1    Emission rates from MAMAP remote sensing data

Remote sensing measurements over the Olinda Alpha Landfill were collected on four different days (27.08.2014, 28.08.2014, 01.09.2014, 03.09.2014) by the MAMAP remote sensing instrument. A detailed list of flight parameters, which were used for the radiative transfer model simulations using SCIATRAN (Rozanov et al., 2014) to generate the look-up table, are found
in Table 1 for each day. For the emission rate estimates, only flight tracks located downwind of the landfill were used. The estimated emission rates as well as the corresponding uncertainties are summarised in Table 2. A detailed error discussion is given in Sects. 3.1.1 and 4.1.1.

     The flight altitude on the four days varied between 1630 and 1970 m asl, the surface elevation was around 300 m asl, the flight speed was around 60 m s$^{-1}$ and the total measurement time per ground sample was around 0.8 s. The ground scene size
for a general flight altitude of around 1800 m asl and this speed in combination with the surface elevation is approximately $69 \times 60$ m$^2$ (cross track $\times$ along track) for a focal length of the installed front optics of $f = 100\,mm$.

     For the remote sensing measurements on the 01.09.2014, the wind direction was estimated to be 241° which is in good agreement with the in-situ based wind direction of 238° derived from in-situ measurements at the plume location of the second downwind wall (dw2 in Fig. 6, a, solid red line; for details of the definition of the plume location, see Sect. 4.2), which was
flown directly after the remote sensing pattern. The wind speed was around 4.4 m s$^{-1}$ determined over the same area as for

**Table 1.** Flight conditions and MAMAP remote sensing parameters for the four flights.

| Flight day | 27.08.2014 | 28.08.2014 | 01.09.2014 | 03.09.2014 |
|---|---|---|---|---|
| Flight time (local time) | | | | |
| start [hh:mm] | 14:11 | 14:21 | 14:55 | 13:27 |
| end [hh:mm] | 14:55 | 15:07 | 16:05 | 14:14 |
| Solar zenith angle (SZA) | | | | |
| min [°] | 29.9 | 31.7 | 38.3 | 27.6 |
| max [°] | 37.0 | 39.3 | 51.3 | 32.6 |
| Flight altitude [m] | 1971 | 1627 | 1794 | 1945 |
| Surface elevation along flight track | | | | |
| min [m] | 80 | 81 | 109 | 114 |
| max [m] | 437 | 435 | 483 | 496 |
| Mean column mixing ratios | | | | |
| $CH_4$ [ppb] | 1748.4 | 1754.1 | 1811.4 | 1799.7 |
| $CO_2$ [ppm] | 398.7 | 397.8 | 393.5 | 394.9 |
| Aerosol scenario [−] | urban | urban | urban | urban |
| Albedo [−] | 0.31 | 0.31 | 0.31 | 0.31 |
| Wind speed [m s$^{-1}$] | 6.3 | 8.1 | 4.4 | 5.5 |
| Wind direction | | | | |
| empirical (center line) [°] | 236 | 240 | 241 | 240 |
| in-situ [°] | 237 | 247 | 238 | 249 |

the wind direction. An overview of the flight pattern and the measured $CH_4$ column enhancements is given in Fig. 6 (a). In addition to a clear plume signal observed up to 8 km downwind of the landfill, some $CH_4$ depletions are visible in the collected data. The origin of these negative $CH_4$ anomalies were investigated in Sect. 3.1.2.

For the emission retrieval, the area between - 1750 and - 4000 m (measurements south of the plume between the yellow lines in Fig. 6, a) and + 1750 and + 4000 m (measurements north of the plume between the yellow lines in Fig. 6, a) was used for background concentration normalization (also compare to Fig. 7, a). The mean emission rate estimate derived from Eq. 1 applied to the 13 downwind tracks (Fig. 7, a) is 13.6 kt $CH_4$ yr$^{-1}$. The corresponding uncertainty is estimated to be 3.8 kt $CH_4$ yr$^{-1}$ (or $\pm 28\%$ of 13.6 kt $CH_4$ yr$^{-1}$).

The MAMAP measurements on 03.09.2014 were treated in a similar way as for the 01.09.2014 flight. The wind direction was 240° based on the empirical center line of the plume (measured in-situ wind direction is 249°). The wind speed was 5.5 m s$^{-1}$. Figure S3 in the supplementary material shows the flight pattern and the $CH_4$ column enhancements.

In order to estimate the emission rate, the data was again filtered by the basic detector filling filter and by inclination. In contrast to the 01.09.2014, the area used for background normalization was set empirically for each track because the flight

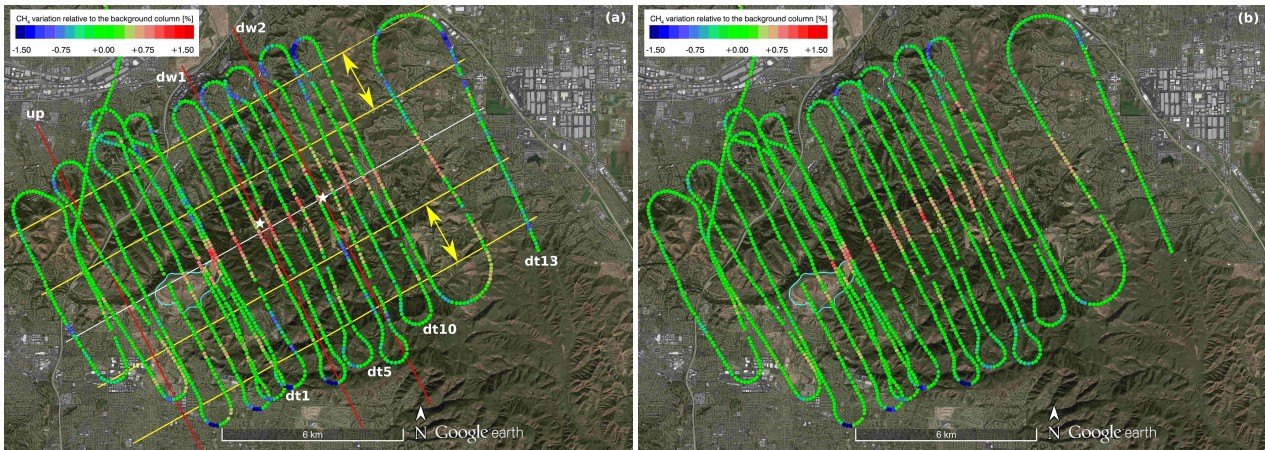

**Figure 6.** The complete MAMAP remote sensing flight pattern without the inclination filter over the Olinda Alpha Landfill (encircled by the cyan solid line) on 01.09.2014 is shown. The $XCH_4(CO_2)$ data is smoothed by a 3-point moving average and normalized by a 300-point moving average for visualisation purpose only. (a) For references, the positions of the center line (solid white line), the normalisation areas (area between the solid yellow lines emphasized by the yellow arrows), the three flown in-situ walls (solid red lines; upwind wall, up; first downwind wall, dw1; second downwind wall, dw2) and labels for the thirteen remote sensing downwind tracks (dt1 to dt13) are also depicted. The white stars emphasize the location of the approximate in-situ plume location, which corresponds to the origin used in Figs. 5, S7 (c-f) and S9 (c,d). (b) Detector filling dependency corrected measurements (for details, see Sect. 3.1.2). The map underneath is provided by Google Earth.

tracks were quite short near the source and longer further away. This was done on basis of the observed plume signal seen in the cross sections (Fig. S4, right column), whereby a broadening of the plume, while moving away from the source, was also considered. Additionally, the maximal width of the plume area of the latter remote sensing tracks was further constrained by the approximate plume width observed in the in-situ measurements. The mean emission based on the 8 downwind tracks is

5    $16.2 \, kt \, CH_4 \, yr^{-1} \, (\pm 23\,\%)$.

The 27.08.2014 and 28.08.2014 flights were more challenging with respect to the flux inversion because of the not optimal flight patterns. This resulted in there being few measurements for concentration background normalisation and a non-optimal orientation of the flight tracks with respect to the prevailing wind direction. Additionally, higher wind speeds potentially led to smaller column enhancements. The flight parameters are listed in Table 1, Figs. S1 and S2 show the flight pattern and Fig. S4

10    (left and middle column) the downwind tracks.

On the 27.08.2014, the area used for background normalization was empirically set and also additionally constrained by the approximate plume width estimated from the in-situ measurements. In contrast to the remaining flights, the inclination filter was relaxed to $6°$ to increase the number of measurements north of the observed plume. Analysis using the 5 downwind tracks yields a mean emission of $13.0 \, kt \, CH_4 \, yr^{-1} \, (\pm 45\,\%)$.

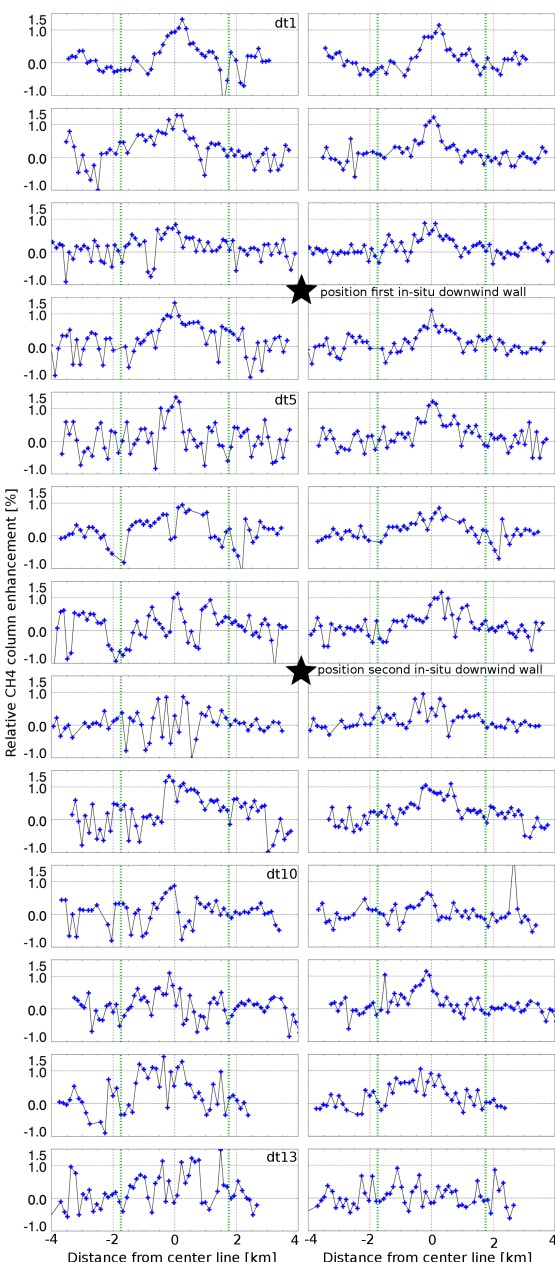

**Figure 7.** Shown are the 13 downwind tracks (filtered for detector filling and inclination, dt1 to dt13) from the MAMAP remote survey over the Olinda Alpha Landfill on 01.09.2014, which were used for the emission rate estimate using Eq. 1. The x-axis depicts the distance from the centre line in km (see also Fig. 6, white solid line) and the y-axis gives the $CH_4$ column enhancement relative to the background column. The area on the left (-4.0 to -1.75 km) and right (+1.75 to +4.0 km) side of the dotted green line was used for background normalisation. Left column: Non-corrected measurements. Right column: Detector filling dependency corrected measurements (see Sect. 3.1.2 for details).

The 28.08.2014 flight was treated in a similar manner to the flights before using again the standard inclination filter of $5°$. The resultant mean emission rate from the 6 downwind tracks is $13.6\,\mathrm{kt\,CH_4\,yr^{-1}}$ ($\pm 39\,\%$).

### 4.1.1 Uncertainties related to remote sensing based emission rates

The uncertainties of the remote sensing based emission rate estimates are based on the methodology described in Sect. 3.1.1 and are listed in Table 2. In the following, a short discussion of the estimated errors for the four measurement flights is given.

**Wind speed ($0.5\,\mathrm{m\,s^{-1}}$):** The resulting uncertainty on the estimated flux is around $\pm 12\,\%$ and $\pm 9\,\%$ for the 01.09.2016 and 03.09.2016 flight. respectively. The uncertainty is slightly smaller on the 27.08.2014 ($\pm 8\,\%$) and on the 27.08.2014 ($\pm 6\,\%$) (compare to Table 1) as a result of the higher wind speeds.

**Wind direction ($10°$):** On the 01.09.2014, the remote sensing tracks were flown nearly perpendicular to the estimated prevailing wind direction with an average deviation of only $3°$. The assumed error in the wind direction of $10°$ leads to an uncertainty in the emission estimate of up to $2\,\%$. For the 03.09.2014 flight, the mean deviation from the perpendicular wind direction was around $13°$ leading to a maximum emission uncertainty of $6\,\%$. The largest mean deviation from the perpendicular wind direction of around $60°$ is observed on the 27.08.2014. For a $\pm 10°$ wind direction uncertainty, this leads to an uncertainty in the emission rate of maximal $22\,\%$. On the 28.09.2014, the deviation of around $35°$ with respect to the perpendicular wind direction was smaller in comparison to the 27.08.2014 flight leading to a maximum uncertainty in the emission rate of $14\,\%$.

**Background normalization area (shifting limits):** Varying and shifting the limits of the background normalisation area yield a maximum change in the emission of around $19\,\%$ and $18\,\%$ for the 01.09.2014 and the 03.09.2014, respectively. For the 27.08.2914 and 28.08.2914, the maximum uncertainty in the emission rate is around $34\,\%$ and $29\,\%$, respectively.

**Track-to-track variability (statistics):** The 1-$\sigma$ track-to-track variability is $\pm 6.8\,\mathrm{kt\,CH_4\,yr^{-1}}$, or $\pm 50\,\%$ of the derived mean emission rate, for a single track and the resulting error on the averaged emission is around $\pm 14\,\%$ when using the 13 downwind tracks on the 01.09.2014. On the 03.09.2014, the observed 1-$\sigma$ uncertainty is $\pm 5.2\,\mathrm{kt\,CH_4\,yr^{-1}}$ (or $\pm 32\,\%$) based on eight tracks yielding an error of around $\pm 11\,\%$ on the mean emission rate. The track-to-track variability is $\pm 4.5\,\mathrm{kt\,CH_4\,yr^{-1}}$ (or $\pm 35\,\%$) on the 27.08.2014 leading to an error on the average of around $\pm 16\,\%$ considering the five downwind tracks. On the 28.08.2014, the track-to-track variability of the six downwind tracks is $\pm 6.2\,\mathrm{kt\,CH_4\,yr^{-1}}$ (or $\pm 46\,\%$) causing an error on the averaged emission rate of around $\pm 19\,\%$.

**Background $CO_2$ variation (proxy method):** Figure 8 shows exemplarily the background normalized IISCs of the two downwind walls on the 01.09.2014 for the background normalized $\mathrm{IISC_{CH_4}}$ (red solid line) and $\mathrm{IISC_{CH_4/CO_2}}$ (blue solid line). On that day, the $CO_2$ plume was co-located to the $CH_4$ plume and causes a reduction of the $CH_4$ plume signal. This finding is consistent with the kriged $CH_4$ and $CO_2$ in-situ measurements in Figs. S7 (d, f for $CH_4$) and S12 (d, f for $CO_2$), which show a well-defined $CO_2$ enhancement at the position of the methane plume. On the 01.09.2014, the derived emission rates are by around $4.6\,\%$ (first downwind wall) to $11.9\,\%$ (second downwind wall) higher if the influence of the $CO_2$ on the emission rate is neglected. Assuming that this in-situ based derived bias is valid for the entire measurement area, which is covered by the remote sensing instruments, indicates that the emission rate estimates based on the remote sensing data are also underestimated by around $4.6\,\%$ to $11.9\,\%$ due to the co-located $CO_2$ on the 01.09.2014. Applying this method to the other downwind walls

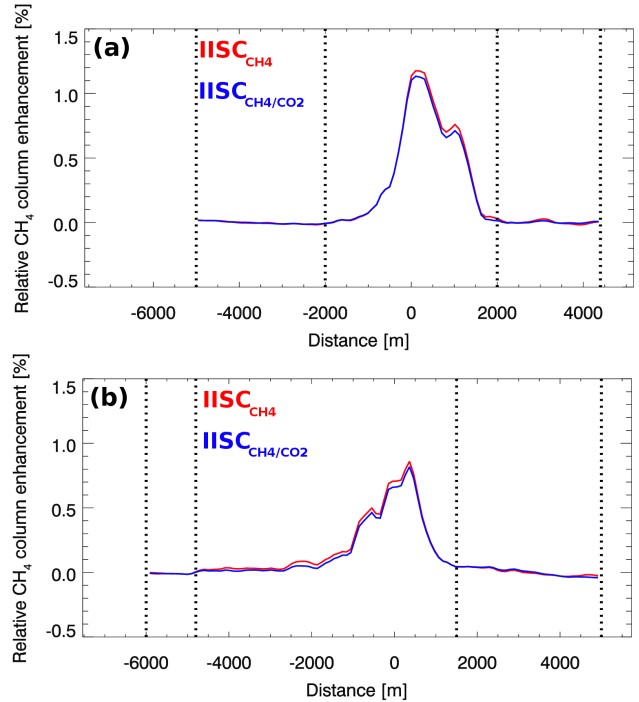

**Figure 8.** Cross-sections of the relative $CH_4$ column enhancements determined from the integrated in-situ columns (IISCs) as discussed in Sect. 3.1.1 of the first (a) and second (b) downwind wall of the Olinda Alpha Landfill measurements on 01.09.2014. The blue solid lines (IISC$_{CH_4/CO_2}$) represent the cases which are influenced by the co-emitted $CO_2$, whereas the red solid lines (IISC$_{CH_4}$) are not. The measurements enclosed by the black dotted lines and located at the flanks / edges of the plume are used for normalization and determination of the background.

yields around $+0.6\%$ (27.08.2014), $-14.9\%$ (28.08.2014) and $+3.3\%$ (03.09.2014). The IISCs of these walls are found in the supplement (Figs. S14, S15 and S16). Strictly speaking, due to the potential temporal and spatial variability of the $CO_2$ variations, these calculated biases estimated from the downwind walls are not assumed to be valid for the remote sensing tracks of the associated flight day, which were recorded at a different time and location. Therefore, we used the 1-$\sigma$ deviation of the derived biases to estimate one uncertainty of around $\pm 10\%$ for the entire remote sensing data set.

**Surface albedo (0.22 and 0.40):** The influence of a wrongly assumed surface albedo used in the RTM simulations has only a minor effect on the estimated emission rates. For the four flights, the relative error is well below $1\%$.

**Total uncertainties:** The resulting total uncertainties including the uncertainties in wind information, normalization area, track-to-track variability, $CO_2$ variations and surface albedo, of the remote sensing measurements for the 01.09.2014, 03.09.2014, 27.06.2014 and 28.06.2014 are $28\%$ (or $3.8\,\mathrm{kt\,CH_4\,yr^{-1}}$), $26\%$ (or $4.2\,\mathrm{kt\,CH_4\,yr^{-1}}$), $45\%$ (or $5.9\,\mathrm{kt\,CH_4\,yr^{-1}}$) and $39\%$ (or $5.3\,\mathrm{kt\,CH_4\,yr^{-1}}$), respectively.

**Table 2.** Summary of the derived emission rates denoted as 'retrieved baseline' and their related relative errors from the remote sensing (RS) and in-situ (IS) data set of the Olinda Alpha Landfill measurements.

| Error type | | 27.08.2014 | 28.08.2014 | 01.09.2014 | | 03.09.2014 |
|---|---|---|---|---|---|---|
| RS | Retrieved baseline [kt $CH_4$ yr$^{-1}$] | 13.0 | 13.6 | 13.6 | | 16.2 |
| | Wind speed [%] | 7.9 | 6.2 | 11.5 | | 9.1 |
| | Wind direction [%] | 22.2 | 13.7 | 2.4 | | 5.5 |
| | Background normalization area [%] | 34.1 | 29.0 | 18.6 | | 18.1 |
| | Track-to-track variability [%] | 15.7 | 18.7 | 13.9 | | 11.4 |
| | Background $CO_2$ variation[a] [%] | 9.9 | 9.9 | 9.9 | | 9.9 |
| | Surface albedo [%] | $<1$ | $<1$ | $<1$ | | $<1$ |
| | Total uncertainty [%] | 45.4 | 38.9 | 27.9 | | 25.9 |
| IS | | dw1[b] | dw1 | dw1 | dw2 | dw1 |
| | Retrieved baseline [kt $CH_4$ yr$^{-1}$] | 11.6 | 16.6 | 17.8 | 14.6 | 13.9 |
| Group a) | Wind speed [%] | 7.9 | 6.2 | 12.5 | 11.5 | 9.1 |
| | Unknown surface concentrations [%] | 6.0 | 8.3 | 17.0 | 9.8 | 13.6 |
| | Time lag [%] | 7.9 | 4.1 | 4.9 | 3.1 | 2.8 |
| Group b) | Kriging parameters [%] | 4.7 | 12.8 | 18.0 | 7.0 | 15.9 |
| | Background concentrations / area [%] | 3.4 | 7.0 | 2.7 | 7.3 | 2.1 |
| | Total uncertainty [%] | 13.9 | 18.4 | 28.3 | 18.4 | 23.1 |

[a] based on the $CH_4$ and $CO_2$ in-situ measurements

[b] dw = downwind wall

## 4.2 Emission rates from Picarro in-situ data

For comparison with the MAMAP remote sensing estimates, $CH_4$ emission rates from the Olinda Alpha Landfill were also derived from consecutive in-situ measurements made by the Picarro instrument performed with the same aircraft for each of the four days, where MAMAP remote sensing data was acquired. In total, five in-situ walls were flown downwind of the landfill during the period. The emission rate estimates for each wall were calculated using the Kriging and mass balance method as described in Sect. 3.2. The downwind walls of the dry gas mixing ratios of $CH_4$ and the effective wind speeds obtained by Kriging can be found in the supplementary material (Sects. S2 and S4).

For the lag separation distance or bin width $h_{sep}$ (see Sect. 3.2), a value of 120 m was chosen for calculating the experimental semivariograms. This value is based on the Picarro instrument, which is the „slowest" in-situ instrument in terms of measurement frequency, whose measurements are used in Eq. 5 for the emission rate estimate. The Picarro greenhouse gas sensor

acquires measurements at around 0.5 Hz, corresponding to a measurement every 2 s. In combination with the flight speed of the aircraft of around $60 \, \mathrm{m \, s^{-1}}$, this leads to a spatial resolution of around 120 m. To cover at least one pair of measurements per lag distance $h_j$, a lag separation distance or bin width $h_{\mathrm{sep}}$ of around 120 m is needed.

For fitting the exponential model to the experimental semivariograms, only half of the maximum possible lag distance (largest distance by which a pair of measurements on the wall is separated) was used following the recommendations in Journel and Huijbregts (1978). Figure 4 shows an example of an experimental semivariogram with the fitted exponential function and the related parameters range, nugget and partial sill.

As mentioned in Sect. 2.3, to account for the fact that measurements were not available at the surface, a pseudo-track was added at the surface. It follows the surface terrain and, in a first order approximation, has the same concentration values of $CH_4$ and $CO_2$ as measured at the altitude of the lowest flight track of the according wall. The surface winds for the pseudo-track were estimated from measurements of the weather station MTNRC1[8] located at the north eastern tip of the Olinda Alpha Landfill. The resulting surface wind speeds and directions at the time the downwind walls were acquired were $5.8 \, \mathrm{m \, s^{-1}}$ and 219° (27.08.2014), $5.9 \, \mathrm{m \, s^{-1}}$ and 228° (28.08.2014), $4.5 \, \mathrm{m \, s^{-1}}$ and 209° (dw1, 01.09.2014), $4.5 \, \mathrm{m \, s^{-1}}$ and 209° (dw2, 01.09.2014), and $4.9 \, \mathrm{m \, s^{-1}}$ and 220° (03.09.2014). This pseudo-track was used to extrapolate the measurements and close the gap between the lowest flight leg and the surface.

The wind speeds for the five downwind walls measured on the four days varied between 4.0 and $8.1 \, \mathrm{m \, s^{-1}}$, retrieved from the measurements by the 5-hole turbulence probe and the surface weather station. These averaged wind speeds were calculated from all grid boxes, which exhibit a $CH_4$ enhancement larger than three times the standard deviation of the $CH_4$ signal in the background area. Subsequently, the wind speeds were also weighted by the amount of the enhanced $CH_4$ molecules in the respective grid boxes. The average area, for which the mean perpendicular wind speeds were calculated over, was around $1.0 \times 1.0 \, \mathrm{km^2}$. This method was chosen to select the wind measurements, which belong to the $CH_4$ plume signal. The 3-$\sigma$ threshold has also been used previously as limit for identifying and distinguishing plume signals from the surrounding background (e.g,. Hörmann et al., 2013; Zien et al., 2014).

The resulting emission rate estimates calculated by Eq. 5 vary between 11.6 and $17.8 \, \mathrm{kt \, CH_4 \, yr^{-1}}$ with corresponding relative uncertainties between 14 % and 28 % during the one week of measurements (see Table 2 for details). When inspecting the three available in-situ upwind walls (Figs. S5, b, S6, b and S7, b), it becomes clear that the calculated emissions are a feature of the emissions from the Olinda Alpha Landfill and are not an artefact of inflow of polluted air masses. The upwind walls do not exhibit any noticeable $CH_4$ enhancements or structures.

### 4.2.1 Uncertainties related to in-situ based emission rate estimates

The error budget for the in-situ based emission rates is shortly discussed in the following. The underlying assumption were presented in Sect. 3.2.1 and uncertainties for the single downwind walls are listed in Table 2.

---

[8]https://www.wunderground.com/personal-weather-station/dashboard?ID=MTNRC1#history, last access: 16.11.2016

**Wind speed ($0.5\,\mathrm{m\,s^{-1}}$):** The averaged absolute wind speeds at the position of the 5 downwind walls varied between 4.0 and $8.1\,\mathrm{m\,s^{-1}}$. This translates into an uncertainty of the estimated emissions of around $6\,\%$ to $13\,\%$ using the accuracy of $0.5\,\mathrm{m\,s^{-1}}$ of the wind probe.

**Unknown surface concentration ($\pm 50\,\%$):** Varying the surface concentrations of $CH_4$ of the pseudo-track at the surface by $\pm 50\,\%$ with respect to the concentrations measured at the lowest flight track, results in emission rate variations between $6\,\%$ and $17\,\%$.

**Time lag ($5\,\mathrm{s}$):** The maximal sensitive of the flux to a changing time lag is between $3\,\%$ and $8\,\%$.

**Kriging parameters:** Varying Kriging parameters for the two quantities wind speed and $CH_4$ concentration have the largest influence on the final emission estimate, whereby the effect of temperature is negligible. The Kriging error results in a flux uncertainty of between $5\,\%$ and $18\,\%$. These tests show, that the influence of the Kriging parameters on the emission is comparable to other error sources but can also be one order of magnitude smaller. It is also important to emphasize, that the chosen values likely reflect the maximum deviations from the derived ones. When inspecting the experimental semivariograms in Fig. 4 it becomes obvious that, e.g., a nugget and partial sill value of $50\,\%$ of the sill or, e.g. in case of $CH_4$, a range reduced to 0.7 or increased to $10.8\,\mathrm{km}$ (fitted value is $2.7\,\mathrm{km}$), respectively, is quite unlikely. Therefore, it is expected, that the real uncertainty originating from the Kriging parameters is smaller.

**Background concentrations / area (shifting limits):** The resulting emission deviates by around $3\,\%$ and $7\,\%$.

**Total uncertainties:** Combining the above mentioned error sources yields total uncertainties of around $14\,\%$ to $28\,\%$ or on average of around $3.1\,\mathrm{kt\,CH_4\,yr^{-1}}$.

### 4.3 $CH_4$ anomaly maps obtained by the AVIRIS-NG instrument

Airborne remote sensing measurements by the AVIRIS-NG imaging spectrometer were performed on the 03.09.2014. The instrument acquired five flight lines over the landfill at an flight altitude of around $3\,\mathrm{km}$ agl between 13:30 and 14:10 local time. The flight lines have a length of approximately $9\,\mathrm{km}$ and a swath of around $1.8\,\mathrm{km}$ resulting in a fine spatial resolution of around $3 \times 3\,m^2$. Figure 9 shows the derived $CH_4$ anomaly map of one flight line in the near field of the landfill using the algorithm described in Sect. 3.3. It shows a clear plume structure developing at the south-western slope of the landfill. This plume is also visible in the $CH_4$ anomaly maps for the remaining AVIRIS-NG overpasses (see supplementary material Fig. S18). Due to atmospheric variability, its shape and intensity changes from overflight to overflight, but the plume remains visible. However, surface structures / surface albedo effects can cause spurious signals, which in the most cases can be identified as such.

### 4.4 Comparison of MAMAP remote sensing with Picarro in-situ data

The estimated emission rates of the Olinda Alpha Landfill from the airborne in-situ and remote sensing measurements agree well for the analysed days (see Fig. 10). Due to the time delay between the two surveys performed with both techniques and, thus, for example a possible change in wind direction, it is not expected that the location of the measured plumes is identical.

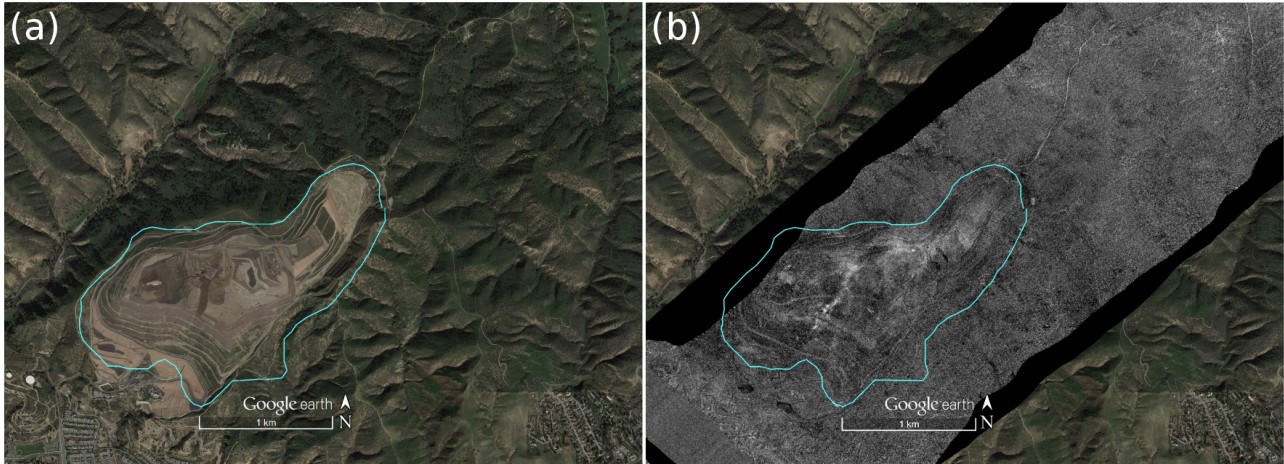

**Figure 9.** (a) Google Earth map showing the Olinda Alpha Landfill encircled by the cyan solid line. (b) $CH_4$ anomaly map derived from AVIRIS-NG data and superimposed on the Google Earth map. Shown is the near field around the landfill acquired at around 13:33 local time. White pixel corresponds to a detected $CH_4$ enhancement. Wind direction was approximately south-west. An enhanced version of the same flight line and view is shown in Fig. 11.

Nevertheless, the positions of the plumes observed by the remote sensing and in-situ instrument are in close vicinity to each other for each of the four days (see Figs. 6, a, S1, S2 and S3).

On the 01.09.2016, the emissions derived from the two in-situ downwind walls are 17.8 kt $CH_4$ yr$^{-1}$ ($\pm 28\,\%$) and 14.6 kt $CH_4$ yr$^{-1}$ ($\pm 18\,\%$), respectively. The difference between the two walls is 3.2 kt $CH_4$ yr$^{-1}$, whereas the average emission rate based on the two in-situ walls is around 16.2 kt $CH_4$ yr$^{-1}$. As suggested in Cambaliza et al. (2014), the difference between the walls can be related to the average emission rate and be used as a measure for the precision of this method. For the flight on the 01.09.2014, this results in a difference of around 20 %, which is in good agreement with the values derived in Cambaliza et al. (2014) ranging from 12 % to 39 %.

Furthermore, the in-situ based emission rates are in good agreement with the remote sensing based emission rates on all four days. The average of the absolute differences between the emission rates based on remote sensing and in-situ is 2.4 kt $CH_4$ yr$^{-1}$. The corresponding uncertainty[9] is 2.8 kt $CH_4$ yr$^{-1}$ pointing out that the in-situ and remote sensing based emission rates are not significantly different.

### 4.5 Qualitative comparison between MAMAP and AVIRIS-NG data

On 03.09.2014, contemporaneous AVIRIS-NG measurements were performed and made available for a qualitative comparison. Figure 11 shows a comparison of the MAMAP remote sensing data on that day with one flight line acquired by AVIRIS-NG at around 13:33 local time. The MAMAP remote sensing measurements were acquired between 13:30 and 14:15 local time.

---

[9]based on error propagation of the single flux uncertainties given in Table 2 and the statistical error

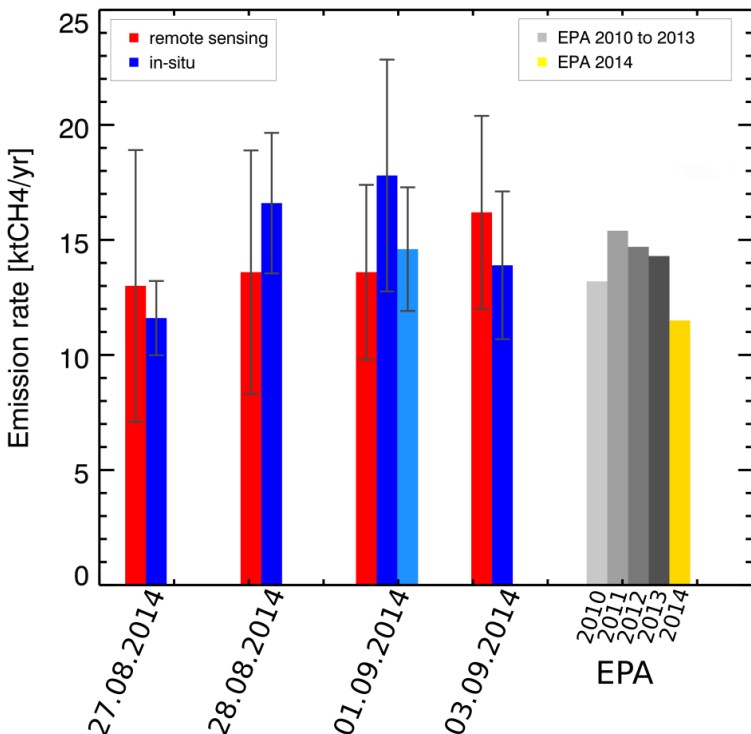

**Figure 10.** The bar charts show the derived emissions and inventory values. The first four sets of bars depict the derived emissions from this study based on the remote sensing (red), in-situ (blue; on the 01.09.2014: Dark blue: Downwind wall 1, bright blue: Downwind wall 2) measurements and their related errors (vertical bars). The fifth set shows the EPA inventory values for the years 2010 to 2013 (grey shaded) and 2014 (yellow).

To better visualize the $CH_4$ plume(s) detected by the AVIRIS-NG instrument on smaller scales, only measurements above a certain threshold are shown in the plot. The AVIRIS-NG data shows a clear plume developing on the south-western slope of the landfill (red arrow) and travelling in downwind direction. It is in good agreement with the $CH_4$ plume seen by the MAMAP instrument.

## 4.6 Comparisons with the EPA inventory

Compared to the EPA inventory value of $11.5\,\mathrm{kt\,CH_4\,yr^{-1}}$ for 2014, our estimated emission rates are on average around $2.8\,\mathrm{kt\,CH_4\,yr^{-1}}$ (with an uncertainty[10] of $\pm 1.6\,\mathrm{kt\,CH_4\,yr^{-1}}$) larger. Due to the scatter of the estimated emission rates and the limited number of measurement days, it is not possible to conclude that EPA is significantly underestimating the Olinda Alpha Landfill $CH_4$ emissions. It is also important to note that the derived fluxes in this work, expressed in units of $\mathrm{kt\,CH_4\,yr^{-1}}$, are only snapshots and valid for the time of the overflight (here: in the afternoon). In addition, the difference could also arise from

---

[10]based on error propagation of the single flux uncertainties given in Table 2 and the statistical error

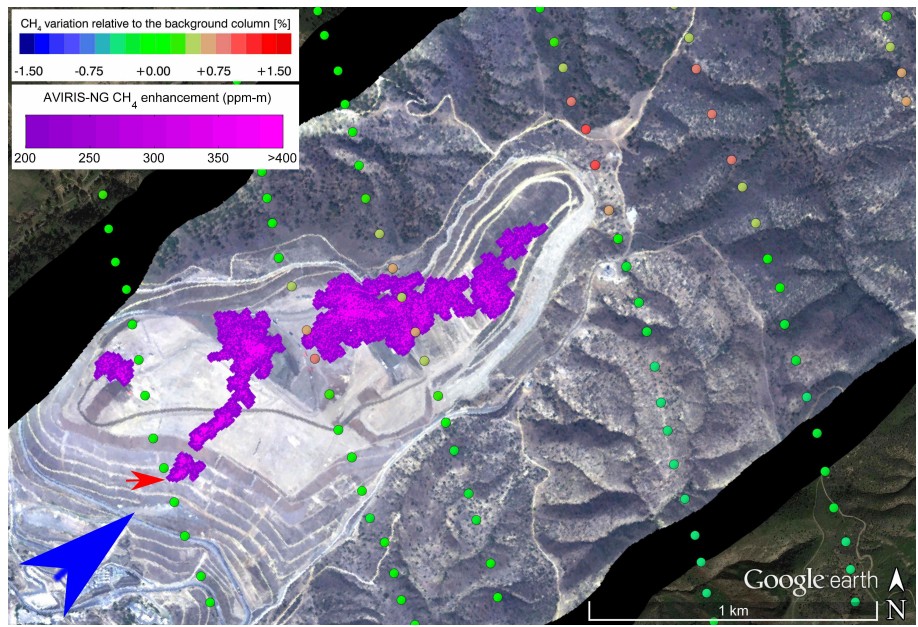

**Figure 11.** The MAMAP remote sensing (coloured circles) and the AVIRIS-NG (pink shaded areas) measurements on the 03.09.2014 are shown. The RGB map underneath is also based on AVIRIS-NG observations. For better source attribution, only AVIRIS-NG measurements having a methane column enhancement of larger than $200\,\mathrm{ppm}\cdot\mathrm{m}$ are shown. The non threshold filtered flight track is depicted in the Fig. 9 (b). The blue arrow depicts the approximate wind direction. Map underneath visible in the upper left and bottom right corner is provided by Google Earth.

the possible leakage identified in the AVIRIS-NG observations, which is not taken into account by EPA, assuming that it was present on all measurement days. Furthermore, e.g., atmospheric pressure variations could potentially also lead to a deviation of the derived fluxes from the inventory value but are difficult to quantify.

## 4.7 Assessment of emission rates of the other measured landfills

Three out of four surveyed landfills (Scholl Canyon Landfill, SCL[11], Puente Hills Landfill, PHL[12] and BKK Landfill[13], compare to Sect. 2.1) did not show well-developed plume structures during the remote sensing survey and, therefore, were not further investigated. In order to assess whether their emission strengths were below the MAMAP remote sensing instrument detection limit for the time of the overflight or whether they were lower than reported, Observation System Simulation Experiments (OSSEs, Gerilowski et al., 2015) have been performed and compared to the actual acquired remote sensing data for the four data sets shown in Fig. 1. The OSSEs are based on Gaussian plume forward model simulations, which incorporates atmospheric conditions like wind speed and wind direction but also considers instrumental characteristics like the MAMAP

---

[11]https://ghgdata.epa.gov/ghgp/service/html/2014?id=1003198&et=undefined, last access: 10.05.2017
[12]https://ghgdata.epa.gov/ghgp/service/html/2014?id=1003199&et=undefined, last access: 10.05.2017
[13]https://ghgdata.epa.gov/ghgp/service/html/2014?id=1011449&et=undefined, last access: 10.05.2017

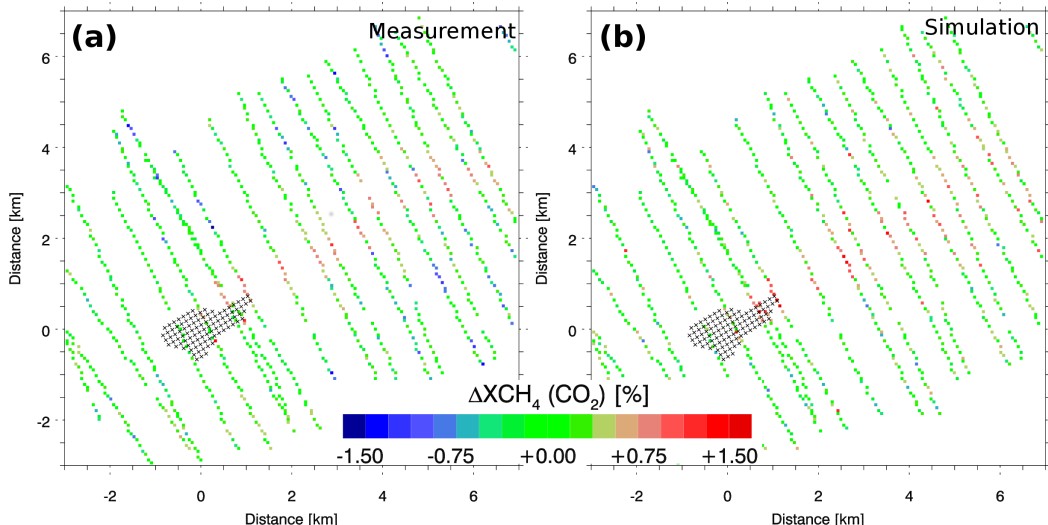

**Figure 12.** Shown are $CH_4$ variations relative to the background column for the Olinda Alpha Landfill in the 01.09.2014. (a) Shows the actual acquired data set similar to that show in Fig. 1 and (b) shows the results from the OSSE. Both data sets are gridded. The crosses represent the sources which were used in the OSSE and are homogeneously distributed across the landfill. The wind direction was south-west. For further details see text and supplementary material Sect. S8.

single measurement precision, the ground scene size and the respective flight track. For the OSSE, multiple sources of equal source strength have been homogeneously distributed across the landfill (for details, see the Sect. S8 in the supplementary material). Fig. 12 (a) shows the MAMAP remote sensing data acquired over the Olinda Alpha Landfill on the 01.09.2014 and Fig. 12 (b) shows the corresponding OSSE, whereby simulated grid points were only plotted if the MAMAP instrument had also

gathered data at the specific positions. In this case, to allow for a better comparison between measurements and simulation, the used emission strength in the OSSE is based on the mean value of the actual measured emission rates on the four days over the Olinda Alpha Landfill and the data have also been gridded to the same grid. There is a good qualitative agreement between simulation and measurements for the Olinda Alpha Landfill on the 01.09.2014 except some blue spots, which have been discussed in Sect. 3.1.2. More details on the OSSEs and the simulations of the three other landfills can be found in the

supplementary material in Sect. S8.

According to EPA, the BKK Landfill is the largest of the four landfills in terms of $CH_4$ emissions. Although its expected $CH_4$ emissions for the year 2014 were around $30\%$ larger than that of the Olinda Alpha Landfill, no enhancements were detected by the MAMAP remote sensing instrument. The measurement flight of the BKK Landfill took place right before the Olinda Alpha Landfill flight on the 01.09.2014. Assuming that the weather conditions were similar for both targets and that BKK was

emitting $15.1\,\mathrm{kt\,CH_4\,yr^{-1}}$, as stated by EPA, the plume should have been detected by the MAMAP remote sensing instrument. This is also confirmed by the OSSE simulations (Fig. S20, d), which shows a clear plume downwind of the landfill for that day assuming an emission rate of $15.1\,\mathrm{kt\,CH_4\,yr^{-1}}$ for the time of the overflight. It is also worthwhile to note that the emission

rate given for the BKK Landfill by EPA might be to high. In case a landfill is equipped with a gas collection system, which is true for all four landfills, the landfill operator needs to report landfill emissions in two different ways (GPO, 2013), whereas the emissions reported by EPA represent always the larger estimate. The first approach A1 relies on forward calculations, whereas the second approach A2 on backward calculations (for details, see Sect. S9 in the supplementary material). In case of OAL both approaches provide similar emission rates (also see Table S2). For BKK, A2 results in emission rates which are 14.2 kt lower than for A1. This large discrepancy may also indicate that the emission of the BKK Landfill are lower than reported.

The reported emissions for the Scholl Canyon Landfill and Puente Hills Landfill are similar for both approaches (2.1 to 5.9 kt $CH_4$ $yr^{-1}$). The OSSE simulation for SCL (Fig. S21, b, in the supplementary material) indicates that these emissions should likely have been visible in the MAMAP remote sensing measurements for the estimated wind conditions. For the PHL, the OSSE simulation (Fig. S21, d, in the supplementary material) indicates that these emissions are below the detection limit of the MAMAP remote sensing instrument for the given days, atmospheric conditions and instrumental characteristics.

## 5 Summary and conclusions

During the COMEX campaign, a comprehensive set of measurements over four landfills located in the Los Angeles Basin were collected. This study analysed in detail the airborne measurements over the most promising target, the Olinda Alpha Landfill, to investigate the use of remote sensing measurements for estimating emission rates of areal sources of around $2\,km^2$ like a landfill. This landfill showed well-developed atmospheric $CH_4$ plume structures on all measurement days, whereas the other three landfills showed no detectable plume structures during the time of the measurements.

The Olinda Alpha Landfill was measured on four days conducted within on week in late summer 2014. During this time period, measurements of column-averaged dry air mole fractions, $XCH_4$, were acquired by the MAMAP remote sensing instrument while flying above the atmospheric boundary layer. In addition, after each remote sensing survey, consecutive in-situ measurements of $CH_4$ and $CO_2$ and other atmospheric parameters like wind speed and wind direction were gathered while probing the atmospheric boundary layer and crossing the plume emitted by the landfill.

Using the collected data set over the Olinda Alpha Landfill, $CH_4$ emission rates have been estimated from the remote sensing data and compared to the emission rates derived from the in-situ measurements. For that, an adapted mass balance approach was used for the emission rate estimates from the remote sensing data. In order to interpret and analyse the in-situ measurements, a Kriging method was applied. The average of the absolute differences between the estimates from both data sets is 2.4 kt $CH_4$ $yr^{-1}$ ($\pm 2.8$ kt $CH_4$ $yr^{-1}$) showing that the estimated emission rates agree well within the errors bars.

The resulting emissions have a range from around 11.6 to 17.8 kt $CH_4$ $yr^{-1}$ with case dependent relative uncertainties of around 14 % to 45 %. The contribution of the different error sources to the total uncertainty varies from case to case. For example, the remote sensing based emission rates are rather sensitive to the chosen background normalization area or number of flight tracks downwind of the landfill. Thus, the uncertainties on the remote sensing based emission rates can be significantly reduced by using better adapted flight patterns for future activities. Additionally, the uncertainty of the remote sensing based

emission rates, which is caused by a not constant $CO_2$ background concentration or by co-emitted $CO_2$ from the landfill, has been estimated by utilizing the $CH_4$ and $CO_2$ in-situ measurements.

In terms of the in-situ measurements, concentration measurements of $CH_4$ at the surface would significantly lower the error in most cases. The error related to the Kriging method used for interpolation between the different flight legs has maximally the same size as other errors but is generally only a minor contributor to the budget. Additionally, it is also based on conservative assumptions.

There is also a good agreement in plume position between the $CH_4$ column enhancements observed by the non-imaging MAMAP instrument and the imaging AVIRIS-NG instrument for data obtained on 03.09.2014. The AVIRIS-NG observations make it possible to identify a $CH_4$ emission hot spot at the slope of the landfill, which could be a potential leakage, e.g., a leak in the cover layer.

Compared to the EPA inventory value, our estimates are on average $2.8\,kt\,CH_4\,yr^{-1}$ ($\pm 1.6\,kt\,CH_4\,yr^{-1}$) higher. This difference might be related to the identified potential leakage not considered by the EPA inventory value or by other reasons e.g., atmospheric pressure variations.

Our study shows for the first time, that medium resolution (FWHM $\approx 0.9\,nm$) airborne based remote sensing measurements in the SWIR region at around $1.65\,\mu m$ are well-suited to estimate total $CH_4$ emissions from landfills at favourable conditions. Observation System Simulation Experiments (OSSEs) have been used to quantitatively investigate the detection limit of the MAMAP remote sensing instrument. The detection limit depends on the prevailing atmospheric conditions as well as on instrumental and flight specific parameters. The reported emission rate of, e.g., the Puente Hills Landfill ($5.0\,kt\,CH_4\,yr^{-1}$) were likely below the MAMAP detection limit at the time of the overflight for the given conditions. For the other landfills, Scholl Canyon ($5.9\,kt\,CH_4\,yr^{-1}$) and BKK Landfill ($15.1\,kt\,CH_4\,yr^{-1}$), the reported emission rates should likely have been visible in the MAMAP remote sensing measurements and, thus, the emission rates might have been smaller than reported.

*Acknowledgements.* The authors declare that they have no conflict of interest. Development of MAMAP was jointly funded by the University and State of Bremen and the Helmholtz Center Potsdam - GFZ German research Centre for Geosciences.

The MAMAP activities within the $CO_2$ and Methane EXperiment (COMEX) were funded in parts by the University and the State of Bremen, the European Space Agency (ESA) and the National Aeronautics and Space Administration (NASA).

Part of the research was carried out at the Jet Propulsion Laboratory, California Institute of Technology, under a contract with the National Aeronautics and Space Administration.

We would like to thank NASA AMES for the support regarding campaign coordination. Especially, we would also like to thank the CIRPAS team for the support during the entire campaign.

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
