# Peer review of "Methane emissions from a Californian landfill, determined from airborne remote sensing and in-situ measurements"

_Atmospheric Measurement Techniques, 2016_

## Referee Comment (RC1) · Anonymous Referee #1 · 6 Feb 2017

Summary: Krautwurst et al. report on a measurement campaign conducted to quantify $CO_2$ and $CH_4$ emissions (and concentration patterns) within the LA Basin. Here, in-situ observations and remote sensing data from the MAMAP instrument and their ability to detect $CH_4$ plumes released from several landfills (and their associated emissions) is discussed. Repeat mass-balance experiments found emissions estimates ranging from 13ktCH4-18.2 ktCH4 with a reported uncertainty range of 17%-46%. While a comparison with another remote sensing instrument revealed qualitative similarities for the observed plume shape.

General comments: The study nicely illustrates a use-case for the MAMAP instrument and how the combined use of airborne remote-sensing and in-situ instruments can help

to assess emissions from a landfill site. Unfortunately, the structure of the paper is confusing and should be revised to help the reader to focus on the key results (see general comments below). For example, the methodology of how uncertainties are calculated (e.g. for $CH_4$ emission rates) is given in the results section (6), while other results are given in the emission comparison section (7). I would suggest to include the comparison with the inventory in the results section. Furthermore, the uncertainty calculation should either be a specific section or logically added to section (5), where the MAMAP retrieval and calculation of emission rates are described. In the last paragraph of the conclusion, the authors claim that this study shows that this type of air-borne remote sensing observations are "well-suited" to estimate $CH_4$ emissions from a "large landfill". Yet, the study showed that for 3 out of 4 landfills investigated, the instrument did NOT detect a significant plume. Here the authors need to critically discuss: Why were the other sites neglected in the analysis and conclusions? What were the EPA emission estimates for the other sites? Maybe there is a detection limit for this methodology or are there other limiting external factors (e.g. meteorology, topography of the site)? These would be crucial information to be added. All that can really be claimed seems to be that, in this instance, a landfill with emissions of above ca. 11ktCH4/a can be monitored using this technique. Concerning the size of the landfill ("large landfill") - I doubt that the size of the landfill is critical here, but rather its $CH_4$ emissions, or maybe its $CH_4$ emission density.

Overall, the study contains important data and very interesting results that could the help the readers to understand GHG emissions at the scale of an industrial site and even better plan future campaigns by e.g. quantifying uncertainty contributions of individual parameters. It has also the potential to gauge the relative value/usefulness of in-situ versus remote-sensing observations in future studies. If the authors can appropriately address the major (and minor) comments this manuscript should be considered for AMT.

Specific comments: P2 Line 7: consider citing a peer-reviewed publication instead of a

webpage here. http://www.earth-syst-sci-data.net/8/697/2016/essd-8-697-2016.pdf

P2 L30f: Please give an order of magnitude for the size. As a significant share of publications in AMT focus on regional to global scale studies, a landfill might not qualify as having a *large* surface area.

P4 L6f: A more diligent reason for why 3 out of 4 landfills are ignored in the manuscript from here on out needs to be given. Why do those other landfills not produce pronounced plumes? What are their (EPA) estimated emissions? Are they lower, equal or higher than the expected emissions at Olinda Alpha? Consider adding a table with the key indicators for all 4 sites.

P4 L9: Why are: "Campaign and target description", "Aircraft instrumentation and collected data sets" and "Flight strategy" three different sections? Please consider combining them as subsections into one "methods" section.

P5 L12: Please correct CDRS to CRDS

P5 L14: What is the uncertainty of total column concentrations determined with the MAMAP instrument? There is a discussion of this in the "results" sections. Please consider moving this discussion into the method section of the paper.

P5 L25: What was the typical uncertainty (repeatability and reproducibility) of the in-situ measurements

P7 L10: Please consider adding the flight track of the second instrument hosting the AVIRIS-NG on Figure 2 if possible.

P9 L2f: The resulting uncertainty of choosing 0.31 as mean albedo could be discussed here. (Similar point for other assumptions/simplifications made throughout section 5).

P10 L3: The authors correctly note that the estimated emissions are only valid during the overflight, yet the units indicate an annual estimate. Suggestion: report the emissions as tCH4 per hour and also calculate the mean hourly emission according to EPA.

This avoids the erroneous implication that annual emission rates can be calculated from this data set.

P14f: After nicely describing all calculations in section 5, why are the uncertainties associated with these calculations now included in the "results" section and not within the previous section? Are the calculated uncertainties considered a key result of this study? If so, this needs to be stated more clearly.

P14/15/16: Here CH4 emissions are reported without any uncertainties and the reader will have to "wait" for the next subsection to judge if emission of 13.0ktCH4/a (27.8.2014) are significantly different from the reported 13.7 ktCH4/a a day later – why?. Please consider restructuring to improve the readability of the manuscript

P18 L6: It seems that only a bias in the determined wind-direction has been accounted for here. What would be the impact of changing wind conditions between two legs of the down-wind legs?

P20 L19: Is subsection 6.1.2 really an independent "result" that needs a subsection or rather additional information about the data exploitation/error calculation?

Table 2: Why is the uncertainty of the albedo of the landfill not considered here?

P23 L25: Why/How was the pseudo-surface concentration enhancement range determined to be 50%-150% of the lowest flight track?

P24 L27: Why is the "comparison of emissions" not considered a result (sect. 6) or included in the conclusions (sect. 8), but discussed in an independent section?

P24 following: Section 7 is called "comparison of emissions" – yet subsection 7.2 compares CH4 concentration results only.

P25 L16: Is the reported average absolute difference statistically significant?

P27/P28: Please consider adding a discussion on the implications of the ability of the suggested observational techniques given that only 1 out of 4 landfills could be studied

within COMEX.

Especially - P28 L28f: Please expand on the claim that the observations are well suited to estimate CH4 emissions from larger landfills (see general comments). The uncertainties of the in-situ estimates you report are smaller than the uncertainties for the remote-sensing estimates and you can only detect stable plumes for 1 out of 4 landfills.

---

## Referee Comment (RC2) · Anonymous Referee #2 · 22 Feb 2017

Krautwurst et al. determined methane emissions from a landfill using both column and in-situ measurements aboard the Twin Otter aircraft. The detailed description of the sampling strategy and the mass balance approach, as well as the analysis of the uncertainties are very useful in general for quantifying methane emissions on a relatively small area, which is significantly larger than point sources. The methodology is sound. It should be considered for publication after taking into account the following comments:

P3/L22: remove "of" after aboard

P4/Figure 1 caption: 27.08.2017 → 27.08.2014

[Figure]

P5/L21: G-2301f is not the flight ready instrument Gxxxx-m, and the aircraft measurements may be affected by ambient pressure change. The mobile version Picarro analyzer have been used in numerous campaigns/publications. Were the Picarro measurements calibrated during flight? The uncertainty of the Picarro measurements should be given, even though it might be a small term compared to the enhancement.

P5/L29: There are several peer-reviewed papers on water corrections that can replace Rella, 2010.

P9/L5-6: what are exactly the vertical profiles from the U.S. standard atmosphere?

P21/L13-16: The spatial resolution of the measurements also depends on the response time of the analyzer? It is therefore important to mention the flow rate of the measurements, and the volume at STP of the cavity to calculate the response time.

P23/L23-24: please also give the range of the deviations according to the assumed enhancement of 50% or 150%.

P27/L2: missing yr-1
* * *

---

## Author Comment (AC1) · 23 Jun 2017

**Reviewer #1 (R#1):**

**First of all, we would like to thank the anonymous reviewer for the detailed and helpful comments. In the following, we first state the comment of the reviewer (R#1-X, except for the 'general comment') and then give our response. References to the manuscript are related to the original discussion manuscript: P2L30 → page 2, line 30.**

General comments: The study nicely illustrates a use-case for the MAMAP instrument and how the combined use of airborne remote-sensing and in-situ instruments can help to assess emissions from a landfill site. Unfortunately, the structure of the paper is confusing and should be revised to help the reader to focus on the key results (see general comments below). For example, the methodology of how uncertainties are calculated (e.g. for CH4 emission rates) is given in the results section (6), while other results are given in the emission comparison section (7). I would suggest to include the comparison with the inventory in the results section. Furthermore, the uncertainty calculation should either be a specific section or logically added to section (5), where the MAMAP retrieval and calculation of emission rates are described. In the last paragraph of the conclusion, the authors claim that this study shows that this type of airborne remote sensing observations are "well-suited" to estimate CH4 emissions from a "large landfill". Yet, the study showed that for 3 out of 4 landfills investigated, the instrument did NOT detect a significant plume. Here the authors need to critically discuss: Why were the other sites neglected in the analysis and conclusions? What were the EPA emission estimates for the other sites? Maybe there is a detection limit for this methodology or are there other limiting external factors (e.g. meteorology, topography of the site)? These would be crucial information to be added. All that can really be claimed seems to be that, in this instance, a landfill with emissions of above ca. 11ktCH4/a can be monitored using this technique. Concerning the size of the landfill ("large landfill") – I doubt that the size of the landfill is critical here, but rather its CH4 emissions, or maybe its CH4 emission density.

**We agree with the reviewer about his/her major concerns regarding the structure of the manuscript and the not existing comprehensive discussion on the three other landfills. We significantly restructured the manuscript – further details are given in the specific responses to R#1-4, R#1-6, R#1-9, R#1-11, R#1-12, R#1-14, R#1-17 and R#1-18. We added a more comprehensive discussion on the three other landfills and also expanded our claims on the suitability of remote sensing measurements for quantifying emission from landfills – further details are given in the specific responses to R#1-3, R#1-20, R#1-21.**

R#1-1) P2 Line 7: consider citing a peer-reviewed publication instead of a webpage here, http://www.earth-syst-sci-data.net/8/697/2016/essd-8-697-2016.pdf

**Done. We updated the reference and replaced Kirschke et al. (2013) by Saunois et al. (2016). Additionally, we also stated the new estimates (P2L7) as given by Saunois et al. (2016) for the year 2012. The final revised version of the manuscript of Saunois et al. (2016) was not published at the time this manuscript (Krautwurst et al., 2016) was written. Therefore we had referenced the older values from Kirschke et al. (2013).**

R#1-2) P2 L30f: Please give an order of magnitude for the size. As a significant share of publications in AMT focus on regional to global scale studies, a landfill might not qualify as having a *large* surface area.

**We agree. At this position in the manuscript, the term 'relatively large' might be confusing for the reader (later in the manuscript, P5L5, the approximate size is stated. In comparison to emissions**

from point sources like power plants or coal mine ventilation shafts, a landfill has a relatively large area from which emissions can potentially develop.) In order to avoid any confusion, we added a size indication 'up to some square kilometres' on P2L30.

R#1-3) P4 L6f: A more diligent reason for why 3 out of 4 landfills are ignored in the manuscript from here on out needs to be given. Why do those other landfills not produce pronounced plumes? What are their (EPA) estimated emissions? Are they lower, equal or higher than the expected emissions at Olinda Alpha? Consider adding a table with the key indicators for all 4 sites.

The aim of this study was not to survey and measure all landfills in the Los Angeles Basin but to show, that it is possible to derive emission rate estimates from remote sensing measurements in the short-wave infrared (SWIR) region. The measurement flights analysed in this manuscript were part of the $CO_2$ and Methane Experiment (COMEX). COMEX was also dedicated to other target types and effects, which were related to the satellite concept missions HyspIRI and CarbonSat. Therefore, the measurement time allocated to landfills was limited. As described in the manuscript (Sect. 4, "Flight strategy"), a target (in this case one of the landfills) was first surveyed by the remote sensing instrument utilising the capabilities of the developed real-time retrieval. Due to the limited amount of flight hours dedicated to landfill measurements, it was decided to gather in-situ measurements within the boundary layer for validation purposes primarily if a plume like structure had already been observed by the remote sensing instrument. As seen in the overview plot (Fig. 1), only the Olinda Alpha Landfill showed pronounced plume like structures in the remote sensing data. In order to show, that remote sensing measurements in the SWIR region have the potential to detect and quantify emissions from landfills, the focus was set to the most promising target, the Olinda Alpha Landfill.

Nevertheless, we agree that the manuscript would benefit from a more comprehensive discussion on the three other landfills, especially, why these landfills do not produce pronounced plumes. Therefore, we added the estimated EPA inventory values for 2014 of all landfills to the main manuscript (Sect. 2 "Campaign and target description", P4L5ff) and also added a more comprehensive overview regarding the reported emissions of the four landfills over the past years and a description of the used methodology how these emissions were estimated in the supplementary material (New Sect. S9, "Landfill reporting practice in the U.S."). Additionally, a further discussion, where we investigated the detection limit of the MAMAP instrument for the other landfills using Observation System Simulation Experiments (OSSEs), was added at the end of the manuscript (see comment to R#1-20).

R#1-4) P4 L9: Why are: "Campaign and target description", "Aircraft instrumentation and collected data sets" and "Flight strategy" three different sections? Please consider combining them as subsections into one "methods" section.

We agree with the reviewer but we would rather call the section "Measurements" instead of "Methods" because the actual methods to retrieve concentrations and emission rates are presented in Sect. 5. Thus, the three sections were summarized by the new section "Measurements".

R#1-5) P5 L12: Please correct CDRS to CRDS

**Done.**

R#1-6) P5 L14: What is the uncertainty of total column concentrations determined with the MAMAP instrument? There is a discussion of this in the "results" sections. Please consider moving this discussion into the method section of the paper.

**In general, the single measurement precision of the retrieved total column concentrations of $CH_4$ (or normalised column-averaged dry air mole fractions of $CH_4$, $XCH_4$) is better than 0.4% (Krings et al., 2013) and was added in Sect. 3 "Aircraft instrumentation and collected data sets" (P5L17). In this context, accuracy is of less importance due to the normalisation of the retrieved columns with the local background beforehand to the flux estimate, see P9L23-25.**

R#1-7) P5 L25: What was the typical uncertainty (repeatability and reproducibility) of the in-situ measurements

**The uncertainty of the Picarro greenhouse gas in-situ analyser has been estimated to around 0.15 ppm for dry $CO_2$ and 2.3 ppb for dry $CH_4$ as determined from laboratory experiments. This uncertainty estimates contain also the contribution due to repeatability of the measurements, whereas by far the largest contribution originates from the water vapour and water vapour correction, respectively. We added the relevant parameters on P5L29.**

R#1-8) P7 L10: Please consider adding the flight track of the second instrument hosting the AVIRIS-NG on Figure 2 if possible.

**This is a good idea, especially for a comparison of the dimensions, but one has to be careful not to clutter Fig. 2 with details and lines, respectively. The second remote sensing instrument AVIRIS-NG aboard a second Twin Otter gathered five flight tracks in total (compare to Sect. S6 in the supplementary material). In order to keep Fig. 2 clear, only one flight track (flight track shown in Fig. S18, c, in the supplementary material and in Figs. 9, b, and 11 in the main manuscript) was added. In case, the reader is interested in the position of another flight track, the landfill is emphasised in Fig. S18 for reference purposes. The approximate dimensions of such a flight line were given in Sect. 6.3 (P24L20).**

R#1-9) P9 L2f: The resulting uncertainty of choosing 0.31 as mean albedo could be discussed here. (Similar point for other assumptions/simplifications made throughout section 5).

**We agree and performed a simple error analysis for the parameter 'surface albedo' to examine its influence on the final emission rate estimate. For that, RTM simulations were performed with the two extreme cases of the surface albedo of 0.22 and 0.40. The resulting effect on the final fluxes was less than 1% for all cases and was added in Table 2.**
**Additionally, we shifted the methodology of how the uncertainties were calculated to Sect. 5 (see also R#1-11).**

R#1-10) P10 L3: The authors correctly note that the estimated emissions are only valid during the overflight, yet the units indicate an annual estimate. Suggestion: report the emissions as tCH4 per hour and also calculate the mean hourly emission according to EPA. This avoids the erroneous implication that annual emission rates can be calculated from this data set.

**We do not agree and consider the unit $ktCH_4/yr$ as more convenient for our purposes. It allows for easy comparison to inventories and to previous publications. As mentioned by the reviewer, we clearly stated in the manuscript that the derived emission rates are strictly speaking only valid for the time of the overflight.**

R#1-11) P14f: After nicely describing all calculations in section 5, why are the uncertainties associated with these calculations now included in the "results" section and not within the previous section? Are the calculated uncertainties considered a key result of this study? If so, this needs to be stated more clearly.

**Agreed. We added the methodology of how errors are calculated to Sects. 5.1 (remote sensing related uncertainties, Sect. 3.1.1 in the revised manuscript) and 5.2 (in-situ related uncertainties, Sect. 3.2.1 in the revised manuscript).**

R#1-12) P14/15/16: Here CH4 emissions are reported without any uncertainties and the reader will have to "wait" for the next subsection to judge if emission of 13.0ktCH4/a (27.8.2014) are significantly different from the reported 13.7 ktCH4/a a day later – why?. Please consider restructuring to improve the readability of the manuscript

**We agree with the reviewer and added the estimated uncertainties to the estimated emission rates (P14L30, P15L10, P16L6, P16L8) in Sects. 6.1 (Sect. 4.1 in the revised manuscript) and 6.2 (Sect. 4.2 in the revised manuscript) to allow for an immediate assessment.**

R#1-13) P18 L6: It seems that only a bias in the determined wind-direction has been accounted for here. What would be the impact of changing wind conditions between two legs of the down-wind legs?

**That is true but we would rather call it maximal uncertainty originating from the wind direction uncertainty of 10°. As stated on P18L6, the wind direction affects the flux via modifying the perpendicular wind speed (via a cosine function).**
**Assuming that all flight tracks of one flight day would be aligned perfectly perpendicular to the prevailing wind direction, the error of the flux due to the wind direction uncertainty of +-10° would then be +-1.5% for each track. One can now assume (as done in the manuscript) that the flux for each track is either wrong by the +1.5% (or -1.5%), which results in a final / total flux also being wrong by +1.5% (or -1.5%).**
**Applying the suggestion from the reviewer would mean that, e.g., for the first track, the wind direction is wrong by +10°, for the second track, the wind direction is wrong by -10°, for the third track, the wind direction is wrong by +10°, etc. In this case, the error on the final / total flux due to an error in the wind direction would be less because the error contributions of the single tracks can cancel.**
**Additionally, if the flight tracks are not aligned perpendicular to the wind direction (standard case), the error due to the wind direction becomes asymmetrical. In this case, always the larger error is given in Table 2.**

R#1-14) P20 L19: Is subsection 6.1.2 really an independent "result" that needs a subsection or rather additional information about the data exploitation/error calculation?

**Subsection 6.1.2 delivers additional information about the data exploitation and, thus, was moved to Sect. 5.1 (Sect. 3.1.2 in the revised manuscript).**

R#1-15) Table 2: Why is the uncertainty of the albedo of the landfill not considered here?

**Uncertainty of the albedo was added to Table 2. See also comment to R#1-9.**

R#1-16) P23 L25: Why/How was the pseudo-surface concentration enhancement range determined to be 50%-150% of the lowest flight track?

**One does actually not know the surface concentrations without measuring them. They can depend on, e.g., the type of source, emission height, atmospheric conditions (stability), and surface terrain.**

**For our baseline, we choose a vertically homogenous distribution of the concentration below the lowest flight track (corresponds to 'pseudo surface track has the same concentrations as measured at the lowest track'). The assumption of a vertically homogenously distributed plume was also chosen in a first order approximation in many studies in case only one flight track and transect, respectively, was available (e.g., Turnbull et al., 2011; Peischl et al., 2013; Karion2013). Additionally, these studies had no measurements at or near the boundary layer height, whereas in our study for the investigated landfill the upper plume limit is well confined.**

**But as shown in Fig. S9, the plume emitted by the landfill cannot be assumed to be well mixed, at least, at the altitudes (usually between 600 and 1200 m) sampled by the aircraft. Therefore, there is no reason to believe that the plume is well-mixed below the lowest flight track. We have tried to capture this effect by the error assessment using, in a first order approximation, a +-50% variation of the $CH_4$ enhancement, which was measured at the lowest flight track, at the surface. In extreme cases, the surface $CH_4$ enhancement could also be -100% or could exceed +100%. We believe that -100% is an unlikely scenario, because landfills are surface sources and $CH_4$ concentrations might increase towards the surface for those source types (Gordon et al., 2015). On the other hand, our measured in-situ walls were acquired some kilometres downwind of the landfill so that it is expected that some vertical mixing had occurred suppressing very high accumulations of $CH_4$ at the surface.**

**The missing measurements of surface concentrations can be a large source of uncertainty in terms of flux calculations. Our error estimate might be on the lower end. This disadvantage can be avoided by using remote sensing measurements, which probe the entire atmospheric column.**

**In order to achieve better estimates of the surface concentrations, one could combine large eddy simulations with the airborne measurements (as done by, e.g., Lavoie et al. 2015) but were out of scope for this work or one could simultaneously collect surface measurements by, e.g., car. We adapted P23L19 – L23 to reflect these thoughts.**

R#1-17) P24 L27: Why is the "comparison of emissions" not considered a result (sect. 6) or included in the conclusions (sect. 8), but discussed in an independent section?

**Agreed. Section "Comparison of emissions" was merged with section "Results".**

R#1-18) P24 following: Section 7 is called "comparison of emissions" – yet subsection 7.2 compares CH4 concentration results only.

**Agreed. Section "Comparison of emissions" was removed and merged with section "Results" (see also comment R#1-17).**

R#1-19) P25 L16: Is the reported average absolute difference statistically significant?

**We changed the term "average absolute difference" to "average of the absolute differences" to not confuse the reader with the also existing term "mean absolute difference (MAD)". We also added the corresponding uncertainty of 2.8 $ktCH_4yr^{-1}$ showing that the average of the absolute differences of 2.4 $ktCH_4yr^{-1}$ is statistically not significant. In this context, we also shortened this section, Sect. 7.1 (Sect. 4.4 in the revised manuscript), to account for better readability. The single emission rates, which had been given in this section, were removed because they can already be found in Table 2 in a more structured and comprehensive way. The average of the absolute differences is a sufficient and meaningful summary of the comparison between the emission rates based on in-situ and remote sensing data.**

R#1-20) P27/P28: Please consider adding a discussion on the implications of the ability of the suggested observational techniques given that only 1 out of 4 landfills could be studied within COMEX.

**We added a dedicated section in the "Results" section (Sect. 4.7 in the revised manuscript) and in the supplementary material (Sect. S8), which investigate possible causes for not measuring $CH_4$ enhancements at the other landfills with the MAMAP remote sensing instrument (also see R#1-3).**

**For that we performed Observation System Simulation Experiments (OSSEs), which are based on Gaussian plume forward model simulations. These simulations incorporate atmospheric conditions like wind speed and wind direction but also consider instrumental characteristics like the MAMAP single measurement precision, ground scene size and the respective flight track. Based on a comparison of simulations and actual measurements it can be investigated whether the reported emission rates by EPA for the three other landfills were below the detection limit of the MAMAP remote sensing instrument or should actually have been detected in the MAMAP measurements for the given days, atmospheric conditions and instrumental and flight characteristics.**

**The results of this investigation have revealed that the reported emission rate for the Puente Hills Landfill (5.0 $ktCH_4yr^{-1}$) might be below the detection limit of the MAMAP remote sensing instrument for the given day, atmospheric conditions and instrumental characteristics.**

**On the other hand, based on the simulations, the emissions of the BKK Landfill (15.1 $ktCH_4yr^{-1}$) and the Scholl Canyon Landfill (5.9 $ktCH_4yr^{-1}$) should likely have been visible in the MAMAP remote**

**sensing measurements. This indicates that the actual emission rates for these landfills for the time of the overflight might have been lower than reported by EPA.**

R#1-21) Especially - P28 L28f: Please expand on the claim that the observations are well suited to estimate CH4 emissions from larger landfills (see general comments). The uncertainties of the in-situ estimates you report are smaller than the uncertainties for the remote-sensing estimates and you can only detect stable plumes for 1 out of 4 landfills.

**We expanded and relativized, respectively, our claim that remote sensing measurements are well suited for measuring landfills based on the previous discussions in R#1-3, R1-20 and R#1-16 (P28L28ff).**

**S8 Observation System Simulation Experiment (OSSE)**

Observation System Simulation Experiments (OSSEs) can be used to simulate the propagation of plumes in the atmosphere originating from various source types and how these plumes would look like if they were measured by, e.g., the MAMAP remote sensing instrument. That means they can be used to qualitatively estimate whether an emission source is observable with the MAMAP remote sensing instrument considering prevailing atmospheric conditions as well as instrumental and flight specific characteristics. The method discussed in the following has been used to estimate, for example, upper-limit emission rates of $CH_4$ for a blowout site located in the North Sea (Gerilowski et. al., 2015) and is based on vertically integrated Gaussian plume forward model simulations (for details, see Krings et. al., 2011 and Gerilowski et al., 2015):

$$V(x,y) = \sum_{i=1}^{N} \frac{F_i}{\sqrt{2\pi}\,\sigma_y(a,x_i)u}\, \exp\left(-\frac{1}{2}\left(\frac{y_i}{\sigma_y(a,x_i)}\right)\right) \qquad \text{Eq. S1}$$

where $V(x,y)$ is the vertically integrated column, which is subsequently normalized by the background column to achieve, e.g., the desired $CH_4$ variation relative to the background column, as a result of one or more emission sources $i$ having emission rates $F_i$, $u$ is the prevailing wind speed which is assumed to be constant across the entire simulation and measurement area, respectively, $\sigma_y(a,x_i)$ is the horizontal dispersion coefficient in across wind direction with the parameter $a$ which depends, in a first order approximation, on wind speed and solar insolation, $x_i$ is the along wind coordinate and $y_i$ the across wind coordinate of source $i$. The sigma sign indicates the summation over all possible sources $i$.

In order to simulate emissions from the landfills under investigation, it was assumed that $CH_4$ emissions took place homogenously distributed across the entire landfill. Therefore, depending on the shape and size of the landfill 90 to 100 single sources were homogenously distributed across the landfill area. Table S1 summarises the parameters necessary for the simulations of the four landfills shown in Fig. 1 in the main manuscript. As these simulations are compared to the actual MAMAP remote sensing measurements, all parameters were derived from the corresponding measurement flights. In order to estimate the wind directions and wind speeds at the BKK Landfill (BKK), Puente Hills Landfill (PHL) and Scholl Canyon Landfill (SCL) sites, we assumed the same vertical wind profile as measured for the Olinda Alpha Landfill (OAL) flight on the corresponding day, but scaled based on a comparison of the surface winds measured by weather stations at the time, the landfills were surveyed. The BKK and PHL are close to the OAL. Therefore, the weather station at the OAL was used to estimate their surface winds. For the SCL, we used the weather station KCAGLEND17 ([https://www.wunderground.com/personal-weather-station/dashboard?ID=KCAGLEND17#history/tgraphs/s20140827/e20140827/mdaily](https://www.wunderground.com/personal-weather-station/dashboard?ID=KCAGLEND17#history/tgraphs/s20140827/e20140827/mdaily)) close to this landfill site. For the simulation of, e.g., the BKK Landfill on the 01.09.2014, the surface wind speed and, thus, the wind speed used for the simulation, was the same as for the subsequent OAL flight (4.4 ms$^{-1}$). The EPA emission rate estimate is 15.1 ktCH$_4$yr$^{-1}$ for 2014, which was equally distributed over the approximate 100 sources. The parameter $a$ used for the horizontal dispersion coefficient $\sigma_y$ is based on the atmospheric stability classification (Turner, 1970) using the wind speed and solar insolation. Thus, a wind speed of 4.4 ms$^{-1}$ and strong solar insolation results in stability class B corresponding to a value of $a$ = 156 (Martin, 1976).

|  | BKK, 01.09.2014 | SCL, 27.08.2014 | PHL, 27.08.2014 | OAL, 01.09.2014 |
|---|---|---|---|---|
| Time of overflight | 14:26 – 14:54 | 11:27 – 12:03 | 12:17 – 13:20 | 14:55 – 16:05 |
| Emission rate [ktCH$_4$yr$^{-1}$] | 15.1 | 5.9 | 5.0 | 14.3 |
| Surface area [km²] | 1.4 | 0.85 | 2.4 | 1.7 |
| Stability class | B | A - B | B | B |
| Parameter a | 156 | 185 | 156 | 156 |
| Wind speed [ms$^{-1}$] | 4.4 | 2.5 | 4.0 | 4.4 |
| Wind direction [°] | 235 | 210 | 227 | 238 |
| Ground scene size [m²] | 69 | 63 | 46 | 64 |
| Precision [%] | 0.30 | 0.27 | 0.33 | 0.34 |

**Table S1: Summery of the relevant quantities used in the Observation System Simulation Experiments (OSSEs).**

Figure S20 (a) shows such a simulation for the BKK Landfill. The resulting column enhancement has also been gridded to pixels having the same size as the approximate ground scene size of the MAMAP remote sensing flight on that day (~ 69 m²) for better comparison. The simulated plume has additionally been rotated in the prevailing wind direction (235°). In the next step (Fig. S20, b), a noise component was added to the simulation to replicate the single measurement precision of the MAMAP remote sensing instrument. The noise was calculated as 1-σ standard deviation (0.30%) from the actual MAMAP remote sensing measurements over the BKK Landfill. In the final step (Fig. d), simulated grid points were only plotted if the MAMAP instrument also gathered data at the specific positions, that is: along the flight track. For comparison, the actual MAMAP flight track over the BKK Landfill on the 01.09.2014 is shown in Fig. S20 (c). From that, one concludes that if the BKK landfill had emitted 15.11 ktCH$_4$yr$^{-1}$ at the time of the measurement, it should likely have been observable by the MAMAP remote sensing instrument (for details, see also main text). The comparison of measurements and simulations for the Scholl Canyon Landfill and Puente Hills Landfill are shown in Fig. S21 and for the Olinda Alpha in Fig. 12 in the main manuscript (Sect. 4.7, including the conclusions from these experiments).

[Figure]

**Figure S 20. (a) Shows the CH₄ variation relative to the background column for the BKK Landfill on the 01.09.2014 based on the OSSE. (b) Shows the same as (a) but with an added noise component. (d) Shows the OSSE only at the position where actual measurement have been acquired. (c) Actual measurement of the MAMAP remote sensing instrument.**

[Figure]

Figure S 21. As Fig. 12 but for the SCL on the 27.08.2014 (a,b) and for the PHL on the 27.08.2014 (c,d).

**S9 Landfill reporting practice in the U.S.**

In the U.S., landfill operators need to report landfill emissions, in case the landfill is equipped with a gas collection system, in two different ways to the United States Environmental Protection Agency (GPO, 2013). The first approach (A1, forward calculation approach) is driven by model data using, e.g., the type and amount of waste, which has historically been deposit within a landfill in combination with a first order decay model. The second approach (A2, back calculation approach) is driven by measurements of the amount of $CH_4$, which has been recovered by the gas collection system, and gas collection efficiencies to estimate $CH_4$ emissions. The official value stated by EPA always represents the larger estimate of the two.

The landfills Olinda Alpha (OAL), BKK, Scholl Canyon (SCL) and Puente Hills (PHL) investigated in this work are equipped with a gas collection system. An overview of the reported emission rates of the four landfills between 2010 and 2015 is given in Table S2.

| | | 2010 | 2011 | 2012 | 2013 | 2014 | 2015 |
|---|---|---|---|---|---|---|---|
| OAL | | | | | | | |
| Emission in $ktCH_4yr^{-1}$ | Official | 13.2 | 15.4 | 14.7 | 14.3 | 11.5 | 12.3 |
| | A1 | 11.2 | 5.9 | 9.3 | 10.4 | 10.0 | 12.3 |
| | A2 | 13.1 | 15.4 | 14.7 | 14.4 | 11.5 | 9.2 |
| BKK | | | | | | | |
| Emission in $ktCH_4yr^{-1}$ | Official | 14.1 | 13.6 | 14.6 | 15.0 | 15.1 | 15.1 |
| | A1 | 14.1 | 13.6 | 14.6 | 15.0 | 15.1 | 15.1 |
| | A2 | 1.3 | 1.3 | 1.2 | 1.0 | 0.9 | 0.9 |
| SCL | | | | | | | |
| Emission in $ktCH_4yr^{-1}$ | Official | 5.6 | 6.9 | 6.5 | 6.3 | 5.9 | 5.3 |
| | A1 | 0.0 | 0.0 | 0.0 | 1.0 | 2.1 | 3.2 |
| | A2 | 5.6 | 6.9 | 6.5 | 6.3 | 5.9 | 5.3 |
| PHL | | | | | | | |
| Emission in $ktCH_4yr^{-1}$ | Official | 17.8 | 17.2 | 17.2 | 10.9 | 5.0 | 13.4 |
| | A1 | 8.3 | 4.1 | 4.2 | 7.7 | 2.4 | 13.3 |
| | A2 | 17.8 | 17.2 | 17.2 | 10.9 | 5.0 | 4.4 |

Table S2: Reported emission rates of the four landfills: Olinda Alpha Landfill (OAL), BKK Landfill (BKK), Scholl Canyon Landfill (SCL), Puente Hills Landfill (PHL). The emission rates for the year 2014 are emphasized in yellow. For each landfill and year three emission rates are given: Official (officially reported by EPA), A1 (forward calculation approach) and A2 (back calculation approach, GPO, 2013).

The emission data for the different facilities and landfills, respectively, from Table S2 can be found at the EPA website:
https://ghgdata.epa.gov/ghgp/main.do, last access: 09.06.2017.
OAL: https://ghgdata.epa.gov/ghgp/service/facilityDetail/2014?id=1002320&ds=E&et=&popup=true, last access: 09.06.2017.
BKK: https://ghgdata.epa.gov/ghgp/service/facilityDetail/2014?id=1011449&ds=E&et=&popup=true, last access: 09.06.2017.
SCL: https://ghgdata.epa.gov/ghgp/service/facilityDetail/2014?id=1003198&ds=E&et=&popup=true, last access: 09.06.2017.
PHL: https://ghgdata.epa.gov/ghgp/service/facilityDetail/2014?id=1003199&ds=E&et=&popup=true, last access: 09.06.2017.

---

## Author Comment (AC2) · 23 Jun 2017

Reviewer #2 (Rf#2):

**First of all, we would like to thank the anonymous reviewer for the detailed and helpful comments. In the following, we first state the comment of the reviewer (Rf#2-X) and then give our response. References to the manuscript are related to the original discussion manuscript: P2L30 → page 2, line 30.**

Krautwurst et al. determined methane emissions from a landfill using both column and in-situ measurements aboard the Twin Otter aircraft. The detailed description of the sampling strategy and the mass balance approach, as well as the analysis of the uncertainties are very useful in general for quantifying methane emissions on a relatively small area, which is significantly larger than point sources. The methodology is sound. It should be considered for publication after taking into account the following comments:

**Please see our detailed response as given below.**

Rf#2-1) P3/L22: remove "of" after aboard

**Done.**

Rf#2-2) P4/Figure 1 caption: 27.08.2017 → 27.08.2014

**Done.**

Rf#2-3) P5/L21: G-2301f is not the flight ready instrument Gxxxx-m, and the aircraft measurements may be affected by ambient pressure change. The mobile version Picarro analyzer have been used in numerous campaigns/publications. Were the Picarro measurements calibrated during flight? The uncertainty of the Picarro measurements should be given, even though it might be a small term compared to the enhancement.

**Flight ready: The flight plan was designed based on previous flight experience of this 2301-f unit. Altitude changes were made not faster than 150 meters per minute, and resulting cavity pressure and temperature were screened against the standard deviations observed by other airborne Picarro instruments (Karion et al. 2013; Johnson et al., 2014). Throughout the data sets analyzed here, cavity pressure had a 1-σ standard deviation of 0.03 – 0.04 Torr, and temperature had a standard deviation of 0.001 – 0.002 °C. Furthermore, no transient behavior in the methane signal was observed to correlate with changes in altitude, as can be seen qualitatively in Fig. 5 (a). The text has been amended to reflect these specific diagnostics (P7L8).**

**Calibration: As the calibration factors are very small, we did not calibrate the data during flight and worked with uncalibrated data in the original manuscript. We investigated the influence of the calibration factors on the results i.e. the emission rate estimates, which was smaller than 1%. For the revised manuscript, we applied the calibration factors by multiplying the dry gas mixing ratios of $CH_4$ and $CO_2$ by the corresponding calibration factors ($CH_4$: 1.002275041, $CO_2$: 1.004664626, factors were added to the manuscript on P5L29). All results, which were affected by the $CH_4$ and $CO_2$ in-situ data, were reanalyzed and accordingly changed in the revised manuscript. As**

mentioned before, those corrections were very small and had no significant effect on the results and conclusions.

**Uncertainty: The uncertainty of the Picarro in-situ measurements of the dry gas mixing ratios of $CO_2$ and $CH_4$ were added to the manuscript (P5L29, see also comment to Rf#1-7 ). They were estimated to be around 0.15 ppm for dry $CO_2$ and 2.3 ppb for dry $CH_4$.**

Rf#2-4) P5/L29: There are several peer-reviewed papers on water corrections that can replace Rella, 2010.

**Agreed. We replaced the White Paper (Rella et al., 2010) by a newer version describing the water vapour correction of the instrument (Rella et al., 2013)**

Rf#2-5) P9/L5-6: what are exactly the vertical profiles from the U.S. standard atmosphere?

**We added the vertical profiles of $CH_4$ and $CO_2$, which were used in retrieval algorithm of the MAMAP remote sensing data, for the different days in the supplementary material (Fig. S19).**

Rf#2-6) P21/L13-16: The spatial resolution of the measurements also depends on the response time of the analyzer? It is therefore important to mention the flow rate of the measurements, and the volume at STP of the cavity to calculate the response time.

**The cell volume $V_{cell}$ of the used analyser is 30 cm³ (working conditions: $p_{cell}$ = 140 Torr and $T_{cell}$ = 318.15 K). Therefore the cell volume $V_{STP}$ at standard temperature and pressure (STP, $p_{STP}$ = 760 Torr and $T_{STP}$ = 273.15 K) is around 4.745 cm³. The flow rate of the system has been analysed in the laboratory. Depending on the ambient pressure and, thus, flight altitude the flow rate slightly varied between 187 (at 600 m above sea mean level, m amsl) and 143 standard cubic centimetres per minute (at 1400 m amsl). This results in refilling times of the cavity of around 1.5 to 2.0 seconds. These values are close to the actual measuring frequency of 0.5 Hz of the Picarro greenhouse gas in-situ analyser. In the original manuscript, we used a wrong measurement frequency of 1.7 Hz. For the revised manuscript, we reanalysed the in-situ data and accordingly changed all emission rate estimates, related uncertainties and figures depending on in-situ data. The resulting changes were only of minor nature, e.g., the final emission rates have changed on average by less than 2% and have had no influence on the final conclusions.**

Rf#2-7) P23/L23-24: please also give the range of the deviations according to the assumed enhancement of 50% or 150%.

**We are not sure whether to understand the question correctly. The assumed deviation concerning a +-50% concentration change at the surface is stated to be +-16% (rounded) at maximum for the flight on the 01.09.2014 (P23L23). For the remaining days, the error due to the surface concentration change is summarised in Table 2 (P22). In Table 2, the absolute emission rates are also given for the single downwind walls so that the estimated error (given in percent) can directly related to the absolute emission rate. Example: Estimated emission rate on 01.09.2014  is 18.2**

ktCH$_4$yr$^{-1}$ for the first downwind wall (dw1), whereas relative uncertainty due to unknown surface concentration is 15.8%, yielding an absolute uncertainty of around 2.9 ktCH$_4$yr$^{-1}$. This uncertainty can be linearly scaled. For example, a CH$_4$ enhancement of +-100% at the surface would correspond to an uncertainty of +- 31.6% or +- 5.8 ktCH$_4$yr$^{-1}$.

Rf#2-8) P27/L2: missing yr-1

**Done.**

**References**

**Johnson, M. S., Yates, E. L., Iraci, L. T., Loewenstein, M., Tadi, J. M., Wecht, K. J., Jeong, S., and Fischer, M. L.: Analyzing source apportioned methane in northern California during Discover-AQ-CA using airborne measurements and model simulations, Atmospheric Environment, 99, 248 – 256, doi:http://dx.doi.org/10.1016/j.atmosenv.2014.09.068, http://www.sciencedirect.com/science/article/pii/S1352231014007614, 2014.**

**Karion, A., Sweeney, C., Wolter, S., Newberger, T., Chen, H., Andrews, A., Kofler, J., Neff, D., and Tans, P.: Long-term greenhouse gas measurements from aircraft, 
[revised manuscript text omitted]
 enhancements can be co-located to the landfill plume for the case when the is co-emitted. This will~~

5

10

15 ~~integrated from the surface to highest altitude of the in-situ wall. Subsequently, the two obtained IISCs for and were similarly treated as they would be in the MAMAP proxy approach. First, the column was divided by the column and then the track was background normalized by its edges. This results on the one hand in an $IISC_{CH_4}$ from the enhancement only, which is not influenced by variations, and on the other in an $IISC_{CH_4/CO_2}$ which considers variations.~~

**Background** $CO_2$ **variation (proxy method):** Figure 8 shows exemplarily the background normalized IISCs of the two

20 downwind walls on the 01.09.2014 for the background normalized $IISC_{CH_4}$ (red solid line) and $IISC_{CH_4/CO_2}$ (blue solid line). On that day, the $CO_2$ plume  was co-located to the $CH_4$ plume and causes a reduction of the $CH_4$ plume signal. This finding is consistent with the kriged $CH_4$ and $CO_2$ in-situ measurements in Figs. S7 (d, f for $CH_4$) and S12 (d, f for $CO_2$), which show a well-defined $CO_2$ enhancement at the position of the methane plume.

25

On the 01.09.2014, the derived emission rates are by around 4.6 % (first downwind wall) to 11.9 % (second downwind wall) higher if the influence of the $CO_2$ on the emission rate is neglected.

30 Assuming that this in-situ based derived bias is valid for the entire measurement area, which is covered by the remote sensing instruments, indicates that the emission rate estimates based on the remote sensing data are also underestimated by around 4.6 % to 11.9 % due to the co-located $CO_2$ on the 01.09.2014.

Applying this method to the other downwind walls yields around +0.6 % (27.08.2014), -14.9 % (28.08.2014) and +3.3 % (03.09.2014). The IISCs of these walls are found in the supplement (Figs. S14, S15 and S16). Strictly speaking, due

35 to the potential temporal and spatial variability of the $CO_2$ variations, these calculated biases estimated from the downwind

walls are not assumed to be valid for the remote sensing tracks of the associated flight day, which were recorded at a different time and location. Therefore, we used the 1-$\sigma$ deviation of the derived biases to estimate one uncertainty of around $\pm 10\%$ for the entire remote sensing data set.

**Surface albedo (0.22 and 0.40):** The influence of a wrongly assumed surface albedo used in the RTM simulations has only a minor effect on the estimated emission rates. For the four flights, the relative error is well below $1\%$.

**Total uncertainties:** The  resulting total uncertainties including the uncertainties in wind information, normalization area, track-to-track variability , $CO_2$ variations and surface albedo, of the remote sensing measurements for the 01.09.2014, 03.09.2014, 27.06.2014 and 28.06.2014 are $28\%$ (or  3.8 kt $CH_4$ $yr^{-1}$), $26\%$ (or  4.2 kt $CH_4$ $yr^{-1}$),  45 % (or 5.9 kt $CH_4$ $yr^{-1}$) and $39\%$ (or 5.3 kt $CH_4$ $yr^{-1}$), respectively.

**4.1.2**

[revised manuscript text omitted]

**S1 MAMAP remote sensing measurments (Google Earth overlays and single tracks)**

[Figure]

**Figure S 1. As Fig. 6 (a) but for the 27.08.2014 and the star corresponds to the origin used in Figs. S5 (c,d) and S9 (a).**

[Figure]

**Figure S 2. As Fig. 6 (a) but for the 28.08.2014 and the star correspond to the origin used in Figs. S6 (c,d) and S9 (b).**

[Figure]

**Figure S 3. As Fig. 6 (a) but for the 03.09.2014 and the star correspond to the origin used in Figs. S8, S9 (c).**

[Figure]

**Figure S 4. As Fig. 7 (left column) but for the three other days (from left to right): 27.08.2014, 28.08.2014 and 03.09.2014.**

**S2 Picarro in-situ dry gas mixing ratis of CH$_4$**

[Figure]

**Figure S 5.** Dry gas mixing ratios of CH$_4$ for the upwind (a,b) and downwind (c,d) wall on 27.08.2014. X-axis gives the distance from the approximate plume centre in m (only for bottom panels) and y-axis gives the altitude in m above mean sea level (m amsl). Solid orange line depicts the surface elevation at the position of the wall (based on SRTM). Dashed black line depicts the area, which was used in the mass balance approach for estimating the emission rate. Dotted black line shows limits, which were used to define the background area (here: from - 3300 to - 2000 m and 2500 to 6500 m). Solid grey line depicts the flight track. (a,c) Measured dry gas mixing ratios of CH$_4$ along the flight track. Each circle represents one measurement. (b,d)  Kriged dry gas mixing ratios of CH$_4$ based on the measurements shown in (a) and (b) and an additionally added pseudo surface track (not shown).

[Figure]

**Figure S 6.** As for Fig. S5 but for the 28.08.2014. (a,b) Upwind wall. (c,d) Downwind wall.

[Figure]

**Figure S 7. As for Fig. S5 but for the 01.09.2014. (a,b) Upwind wall. (c,d) First downwind wall. (e,f) Second downwind wall.**

[Figure]

**Figure S 8. As for Fig. S5 but for the 03.01.2014. (a,b) Downwind wall.**

[Figure]

**Figure S 9.** Shown are enhanced dry gas mixing ratios of CH₄ of the five downwind walls acquired on the four different flight days 27.08.2014 (a), 28.08.2014 (b), 01.09.2014 (c, first downwind wall; d, second downwind wall) and 03.09.2014 (e). Only the area, which was used in the mass balance approach, is shown (dashed black line). Solid orange line depicts the surface elevation at the position of the wall (based on SRTM). Solid grey line shows the flight track.

**S3 Picarro in-situ dry gas mixing ratios of $CO_2$**

[Figure]

**Figure S 10. As Fig. S5 but for the dry gas mixing ratios of $CO_2$ on 27.08.2014 and without dashed and dotted lines. (a,b) Upwind wall. (c,d) Downwind wall.**

[Figure]

**Figure S 11. As for Fig. S10 but for the 28.08.2014. (a,b) Upwind wall. (c,d) Downwind wall.**

[Figure]

**Figure S 12.** As for Fig. S10 but for the 01.09.2014. (a,b) Upwind wall. (c,d) First downwind wall. (e,f) Second downwind wall.

[Figure]

**Figure S 13.** As for Fig. S10 but for the 03.01.2014. (a,b) Downwind wall.

**S4 Horizontal wind fields u$_{eff}$ used in the mass balance approach**

[Figure]

**Figure S 14.** Shown are the horizontal wind fields u$_{eff}$ of the five downwind walls used in the mass balance approach acquired on the four flight days 27.08.2014 (a,b), 28.08.2014 (c,d), 01.09.2014 (e,f, first downwind wall; g,h, second downwind wall) and 03.09.2014 (i,j). Measurements are filtered by an inclination of 5° (see also main text). The area used in the mass balance approach is bordered by a dashed black line. Dotted black line shows limits, which were used to define the CH$_4$ background area. Solid orange line depicts the surface elevation at the position of the wall (based on SRTM). Solid grey line shows the flight track.

**S5 Integrated in-situ columns (IISC)**

[Figure]

**Figure S 15.** Ratios of the integrated in-situ columns of $CH_4$ and $CO_2$ for the upwind walls (a,c) and downwind walls (b,d) on the 27.08.2014 (a,b) and 28.08.2014 (c,d). The measurements enclosed by the black dotted lines and located at the flanks / edges of the plume are used for normalization (compare to Fig. S17).

[Figure]

**Figure S 16.** As Fig. S15 but for the upwind wall (a), first downwind wall (b) and second downwind wall (c) on the 01.09.2014 and the downwind wall (d) on the 03.09.2014 (also compare to Fig. S17).

[Figure]

Figure S 17. As Fig . 7 but for all downwind walls. (a) 27.08.2014. (b) 28.08.2014. (c) 01.09.2014, first downwind wall (as in Fig. 7, a). (d) 03.09.2014, second downwind wall (as in Fig. 7, b). (e) 03.09.2014.

**S6 AVIRIS-NG CH$_4$ retrieval results (Google Earth overlays)**

[Figure]

**Figure S 18. Overview of the methane retrieval results from the AVIRIS-NG observations from different overflight times [local time]: a) Underlying Google Earth Map of the Olinda Alpha Landfill which is emphasized by the cyan solid line. (b) 13:31. (c) 13:33, same overflight as shown in Fig. 9 in the main part. (d) 13:38. (e) 13:48. (f) 14:06.**

**S7 Vertical background profiles of CO$_2$ and CH$_4$**

[Figure]

Figure S 19. Shown are the scaled background profiles (as described in Sect. 3.1 of the main manuscript), which were used in the MAMAP remote sensing retrieval: (a) 27.08.2014. (b) 28.08.2014. (c) 01.09.2014. (d) 03.09.2014

**S8 Observation System Simulation Experiment (OSSE)**

Observation System Simulation Experiments (OSSEs) can be used to simulate the propagation of plumes in the atmosphere originating from various source types and how these plumes would look like if they were measured by, e.g., the MAMAP remote sensing instrument. That means they can be used to qualitatively estimate whether an emission source is observable with the MAMAP remote sensing instrument considering prevailing atmospheric conditions as well as instrumental and flight specific characteristics. The method discussed in the following has been used to estimate, for example, upper-limit emission rates of $CH_4$ for a blowout site located in the North Sea (Gerilowski et. al., 2015) and is based on vertically integrated Gaussian plume forward model simulations (for details, see Krings et. al., 2011 and Gerilowski et al., 2015):

$$V(x,y) = \sum_{i=1}^{N} \frac{F_i}{\sqrt{2\pi}\, \sigma_y(a,x_i)u}\, \exp\left(-\frac{1}{2}\left(\frac{y_i}{\sigma_y(a,x_i)}\right)\right) \qquad \text{Eq. S1}$$

where $V(x,y)$ is the vertically integrated column, which is subsequently normalized by the background column to achieve, e.g., the desired $CH_4$ variation relative to the background column, as a result of one or more emission sources $i$ having emission rates $F_i$, $u$ is the prevailing wind speed which is assumed to be constant across the entire simulation and measurement area, respectively, $\sigma_y(a,x_i)$ is the horizontal dispersion coefficient in across wind direction with the parameter $a$ which depends, in a first order approximation, on wind speed and solar insolation, $x_i$ is the along wind coordinate and $y_i$ the across wind coordinate of source $i$. The sigma sign indicates the summation over all possible sources $i$.

In order to simulate emissions from the landfills under investigation, it was assumed that $CH_4$ emissions took place homogenously distributed across the entire landfill. Therefore, depending on the shape and size of the landfill 90 to 100 single sources were homogenously distributed across the landfill area. Table S1 summarises the parameters necessary for the simulations of the four landfills shown in Fig. 1 in the main manuscript. As these simulations are compared to the actual MAMAP remote sensing measurements, all parameters were derived from the corresponding measurement flights. In order to estimate the wind directions and wind speeds at the BKK Landfill (BKK), Puente Hills Landfill (PHL) and Scholl Canyon Landfill (SCL) sites, we assumed the same vertical wind profile as measured for the Olinda Alpha Landfill (OAL) flight on the corresponding day, but scaled based on a comparison of the surface winds measured by weather stations at the time, the landfills were surveyed. The BKK and PHL are close to the OAL. Therefore, the weather station at the OAL was used to estimate their surface winds. For the SCL, we used the weather station KCAGLEND17 (https://www.wunderground.com/personal-weather-station/dashboard?ID=KCAGLEND17#history/tgraphs/s20140827/e20140827/mdaily) close to this landfill site. For the simulation of, e.g., the BKK Landfill on the 01.09.2014, the surface wind speed and, thus, the wind speed used for the simulation, was the same as for the subsequent OAL flight (4.4 ms[-1]). The EPA emission rate estimate is 15.1 ktCH$_4$yr[-1] for 2014, which was equally distributed over the approximate 100 sources. The parameter $a$ used for the horizontal dispersion coefficient $\sigma_y$ is based on the atmospheric stability classification (Turner, 1970) using the wind speed and solar insolation. Thus, a wind speed of 4.4 ms[-1] and strong solar insolation results in stability class B corresponding to a value of $a$ = 156 (Martin, 1976).

|  | BKK, 01.09.2014 | SCL, 27.08.2014 | PHL, 27.08.2014 | OAL, 01.09.2014 |
|---|---|---|---|---|
| Time of overflight | 14:26 – 14:54 | 11:27 – 12:03 | 12:17 – 13:20 | 14:55 – 16:05 |
| Emission rate [ktCH$_4$yr$^{-1}$] | 15.1 | 5.9 | 5.0 | 14.3 |
| Surface area [km²] | 1.4 | 0.85 | 2.4 | 1.7 |
| Stability class | B | A - B | B | B |
| Parameter a | 156 | 185 | 156 | 156 |
| Wind speed [ms$^{-1}$] | 4.4 | 2.5 | 4.0 | 4.4 |
| Wind direction [°] | 235 | 210 | 227 | 238 |
| Ground scene size [m²] | 69 | 63 | 46 | 64 |
| Precision [%] | 0.30 | 0.27 | 0.33 | 0.34 |

**Table S1: Summery of the relevant quantities used in the Observation System Simulation Experiments (OSSEs).**

Figure S20 (a) shows such a simulation for the BKK Landfill. The resulting column enhancement has also been gridded to pixels having the same size as the approximate ground scene size of the MAMAP remote sensing flight on that day (~ 69 m²) for better comparison. The simulated plume has additionally been rotated in the prevailing wind direction (235°). In the next step (Fig. S20, b), a noise component was added to the simulation to replicate the single measurement precision of the MAMAP remote sensing instrument. The noise was calculated as 1-σ standard deviation (0.30%) from the actual MAMAP remote sensing measurements over the BKK Landfill. In the final step (Fig. d), simulated grid points were only plotted if the MAMAP instrument also gathered data at the specific positions, that is: along the flight track. For comparison, the actual MAMAP flight track over the BKK Landfill on the 01.09.2014 is shown in Fig. S20 (c). From that, one concludes that if the BKK landfill had emitted 15.11 ktCH$_4$yr$^{-1}$ at the time of the measurement, it should likely have been observable by the MAMAP remote sensing instrument (for details, see also main text). The comparison of measurements and simulations for the Scholl Canyon Landfill and Puente Hills Landfill are shown in Fig. S21 and for the Olinda Alpha in Fig. 12 in the main manuscript (Sect. 4.7, including the conclusions from these experiments).

[Figure]

**Figure S 20. (a) Shows the CH₄ variation relative to the background column for the BKK Landfill on the 01.09.2014 based on the OSSE. (b) Shows the same as (a) but with an added noise component. (d) Shows the OSSE only at the position where actual measurement have been acquired. (c) Actual measurement of the MAMAP remote sensing instrument.**

[Figure]

Figure S 21. As Fig. 12 but for the SCL on the 27.08.2014 (a,b) and for the PHL on the 27.08.2014 (c,d).

**S9 Landfill reporting practice in the U.S.**

In the U.S., landfill operators need to report landfill emissions, in case the landfill is equipped with a gas collection system, in two different ways to the United States Environmental Protection Agency (GPO, 2013). The first approach (A1, forward calculation approach) is driven by model data using, e.g., the type and amount of waste, which has historically been deposit within a landfill in combination with a first order decay model. The second approach (A2, back calculation approach) is driven by measurements of the amount of $CH_4$, which has been recovered by the gas collection system, and gas collection efficiencies to estimate $CH_4$ emissions. The official value stated by EPA always represents the larger estimate of the two.

The landfills Olinda Alpha (OAL), BKK, Scholl Canyon (SCL) and Puente Hills (PHL) investigated in this work are equipped with a gas collection system. An overview of the reported emission rates of the four landfills between 2010 and 2015 is given in Table S2.

| | | 2010 | 2011 | 2012 | 2013 | 2014 | 2015 |
|---|---|---|---|---|---|---|---|
| OAL | | | | | | | |
| Emission in $ktCH_4yr^{-1}$ | Official | 13.2 | 15.4 | 14.7 | 14.3 | 11.5 | 12.3 |
| | A1 | 11.2 | 5.9 | 9.3 | 10.4 | 10.0 | 12.3 |
| | A2 | 13.1 | 15.4 | 14.7 | 14.4 | 11.5 | 9.2 |
| BKK | | | | | | | |
| Emission in $ktCH_4yr^{-1}$ | Official | 14.1 | 13.6 | 14.6 | 15.0 | 15.1 | 15.1 |
| | A1 | 14.1 | 13.6 | 14.6 | 15.0 | 15.1 | 15.1 |
| | A2 | 1.3 | 1.3 | 1.2 | 1.0 | 0.9 | 0.9 |
| SCL | | | | | | | |
| Emission in $ktCH_4yr^{-1}$ | Official | 5.6 | 6.9 | 6.5 | 6.3 | 5.9 | 5.3 |
| | A1 | 0.0 | 0.0 | 0.0 | 1.0 | 2.1 | 3.2 |
| | A2 | 5.6 | 6.9 | 6.5 | 6.3 | 5.9 | 5.3 |
| PHL | | | | | | | |
| Emission in $ktCH_4yr^{-1}$ | Official | 17.8 | 17.2 | 17.2 | 10.9 | 5.0 | 13.4 |
| | A1 | 8.3 | 4.1 | 4.2 | 7.7 | 2.4 | 13.3 |
| | A2 | 17.8 | 17.2 | 17.2 | 10.9 | 5.0 | 4.4 |

Table S2: Reported emission rates of the four landfills: Olinda Alpha Landfill (OAL), BKK Landfill (BKK), Scholl Canyon Landfill (SCL), Puente Hills Landfill (PHL). The emission rates for the year 2014 are emphasized in yellow. For each landfill and year three emission rates are given: Official (officially reported by EPA), A1 (forward calculation approach) and A2 (back calculation approach, GPO, 2013).

The emission data for the different facilities and landfills, respectively, from Table S2 can be found at the EPA website:
https://ghgdata.epa.gov/ghgp/main.do, last access: 09.06.2017.
OAL: https://ghgdata.epa.gov/ghgp/service/facilityDetail/2014?id=1002320&ds=E&et=&popup=true, last access: 09.06.2017.
BKK: https://ghgdata.epa.gov/ghgp/service/facilityDetail/2014?id=1011449&ds=E&et=&popup=true, last access: 09.06.2017.
SCL: https://ghgdata.epa.gov/ghgp/service/facilityDetail/2014?id=1003198&ds=E&et=&popup=true, last access: 09.06.2017.
PHL: https://ghgdata.epa.gov/ghgp/service/facilityDetail/2014?id=1003199&ds=E&et=&popup=true, last access: 09.06.2017.